# NAS-𝒳: Neural Adaptive Smoothing via Twisting

**Dieterich Lawson**[*,†]
Google Research
dieterichl@google.com

**Michael Y. Li**[*]
Stanford University
michaelyli@stanford.edu

**Scott W. Linderman**
Stanford University
scott.linderman@stanford.edu

## Abstract

Sequential latent variable models (SLVMs) are essential tools in statistics and machine learning, with applications ranging from healthcare to neuroscience. As their flexibility increases, analytic inference and model learning can become challenging, necessitating approximate methods. Here we introduce neural adaptive smoothing via twisting (NAS-X), a method that extends reweighted wake-sleep (RWS) to the sequential setting by using smoothing sequential Monte Carlo (SMC) to estimate intractable posterior expectations. Combining RWS and smoothing SMC allows NAS-X to provide low-bias and low-variance gradient estimates, and fit both discrete and continuous latent variable models. We illustrate the theoretical advantages of NAS-X over previous methods and explore these advantages empirically in a variety of tasks, including a challenging application to mechanistic models of neuronal dynamics. These experiments show that NAS-X substantially outperforms previous VI- and RWS-based methods in inference and model learning, achieving lower parameter error and tighter likelihood bounds.

## 1 Introduction

Sequential latent variable models (SLVMs) are a foundational model class in statistics and machine learning, propelled by the success of hidden Markov models [1] and linear dynamical systems [2]. To model more complex data, SLVMs have incorporated nonlinear conditional dependencies, resulting in models such as sequential variational autoencoders [3–7], financial volatility models [8], and biophysical models of neural activity [9]. While these nonlinear dependencies make SLVMs more flexible, they also frustrate inference and model learning, motivating the search for approximate methods.

One popular method for inference in nonlinear SLVMs is sequential Monte Carlo (SMC), which provides a weighted particle approximation to the true posterior and an unbiased estimator of the marginal likelihood. For most SLVMs, SMC is a significant improvement over standard importance sampling, providing an estimator of the marginal likelihood with variance that grows linearly in the length of the sequence rather than exponentially [10]. SMC's performance, however, depends on having a suitable proposal. The optimal proposal is often intractable, so in practice, proposal parameters are learned from data. Broadly, there are two approaches to proposal learning: variational inference [11, 12] and reweighted wake-sleep [13, 14].

---

[*] Equal contribution. [†] Work performed while at Stanford University.

37th Conference on Neural Information Processing Systems (NeurIPS 2023).

Variational inference (VI) methods for SLVMs optimize the model and proposal parameters by ascending a lower bound on the log marginal likelihood that can be estimated with SMC, an approach called variational SMC [15–17]. Recent advances in variational SMC have extended it to work with smoothing SMC [18, 19], a crucial development for models where future observations are strongly related to previous model states. Without smoothing SMC, particle degeneracy can cause high variance of the lower bound and its gradients, resulting in unstable learning [18, 10].

Reweighted wake-sleep (RWS) methods instead use SMC's posterior approximation to directly estimate gradients of the log marginal likelihood [14, 20]. This approach can be interpreted as descending the inclusive KL divergence from the true posterior to the proposal. Notably, RWS methods work with discrete latent variables, providing a compelling advantage over VI methods that typically resort to high-variance score function estimates for discrete latent variables.

In this work, we combine recent advances in smoothing variational SMC with the benefits of reweighted wake-sleep. To this end, we introduce neural adaptive smoothing via twisting (NAS-X), a method for inference and model learning in nonlinear SLVMs that uses smoothing SMC to approximate intractable posterior expectations. The result is a versatile, low-bias and low-variance estimator of the gradients of the log marginal likelihood suitable for fitting proposal and model parameters in both continuous and discrete SLVMs.

After introducing NAS-X, we present two theoretical results that highlight the advantages of NAS-X over other RWS-based methods. We also demonstrate NAS-X's performance empirically in model learning and inference in linear Gaussian state-space models, discrete latent variable models, and high-dimensional ODE-based mechanistic models of neural dynamics. In all experiments, we find that NAS-X substantially outperforms several VI and RWS alternatives including ELBO, IWAE, FIVO, SIXO. Furthermore, we empirically show that our method enjoys lower bias and lower variance gradients, requires minimal additional computational overhead, and is robust and easy to train.

## 2 Background

This work considers model learning and inference in nonlinear sequential latent variable models with Markovian structure; i.e. models that factor as

$$p_{\boldsymbol{\theta}}(\mathbf{x}_{1:T}, \mathbf{y}_{1:T}) = p_{\boldsymbol{\theta}}(\mathbf{x}_1)p_{\boldsymbol{\theta}}(\mathbf{y}_1 \mid \mathbf{x}_1) \prod_{t=2}^{T} p_{\boldsymbol{\theta}}(\mathbf{x}_t \mid \mathbf{x}_{t-1})p_{\boldsymbol{\theta}}(\mathbf{y}_t \mid \mathbf{x}_t), \tag{1}$$

with latent variables $\mathbf{x}_{1:T} \in \mathcal{X}^T$, observations $\mathbf{y}_{1:T} \in \mathcal{Y}^T$, and global parameters $\boldsymbol{\theta} \in \Theta$. By nonlinear, we mean latent variable models where the parameters of the conditional distributions $p_{\boldsymbol{\theta}}(\mathbf{x}_t \mid \mathbf{x}_{t-1})$ and $p_{\boldsymbol{\theta}}(\mathbf{y}_t \mid \mathbf{x}_t)$ depend nonlinearly on $\mathbf{x}_{t-1}$ and $\mathbf{x}_t$, respectively.

Estimating the marginal likelihood $p_{\theta}(\mathbf{y}_{1:T})$ and posterior $p_{\theta}(\mathbf{x}_{1:T} \mid \mathbf{y}_{1:T})$ for this model class is difficult because it requires computing an intractable integral over the latents,

$$p_{\boldsymbol{\theta}}(\mathbf{y}_{1:T}) = \int_{\mathcal{X}^T} p_{\boldsymbol{\theta}}(\mathbf{y}_{1:T}, \mathbf{x}_{1:T}) \, \mathrm{d}\mathbf{x}_{1:T},$$

We begin by introducing two algorithms, reweighted wake-sleep [14, 13] and smoothing sequential Monte Carlo [19], that are crucial for understanding our approach.

### 2.1 Reweighted Wake-Sleep

Reweighted wake-sleep (RWS, [13, 14]) is a method for maximum marginal likelihood in LVMs that estimates the gradients of the marginal likelihood using self-normalized importance sampling. This is motivated by Fisher's identity, which allows us to write the gradients of the marginal likelihood as a posterior expectation,

$$\nabla_\theta \log p_\theta(\mathbf{y}_{1:T}) = \mathbb{E}_{p_\theta(\mathbf{x}_{1:T} \mid \mathbf{y}_{1:T})} \left[ \nabla_\theta \log p_\theta(\mathbf{x}_{1:T}, \mathbf{y}_{1:T}) \right], \tag{2}$$

as proved in Appendix 8.1. The term inside the expectation is computable with modern automatic differentiation tools [21, 22], but the posterior $p_\theta(\mathbf{x}_{1:T} \mid \mathbf{y}_{1:T})$ is unavailable. Thus, SNIS is used to form a biased but consistent Monte Carlo estimate of Eq. (2) [23]. Specifically, SNIS draws $N$ IID

samples from a proposal distribution, $q_\phi(\mathbf{x}_{1:T} \mid \mathbf{y}_{1:T})$ and weights them to form the estimator

$$\sum_{i=1}^{N} \overline{w}^{(i)} \nabla_\theta \log p_\theta(\mathbf{x}_{1:T}^{(i)}, \mathbf{y}_{1:T}), \quad \mathbf{x}_{1:T}^{(i)} \sim q_\phi(\mathbf{x}_{1:T} \mid \mathbf{y}_{1:T}), \quad \overline{w}^{(i)} \propto \frac{p_\theta(\mathbf{x}_{1:T}^{(i)}, \mathbf{y}_{1:T})}{q_\phi(\mathbf{x}_{1:T}^{(i)} \mid \mathbf{y}_{1:T})} \quad (3)$$

where $\overline{w}^{(i)}$ are normalized weights, i.e. $\sum_{i=1}^{N} \overline{w}^{(i)} = 1$.

The variance of this estimator is reduced as $q_\phi$ approaches the posterior [23], so RWS also updates $q_\phi$ by minimizing the inclusive Kullback-Leibler (KL) divergence from the posterior to the proposal. Crucially, the gradient for this step can also be written as the posterior expectation

$$\nabla_\phi \mathrm{KL}(p_\theta(\mathbf{x}_{1:T} \mid \mathbf{y}_{1:T}) \mid\mid q_\phi(\mathbf{x}_{1:T} \mid \mathbf{y}_{1:T})) = -\mathbb{E}_{p_\theta(\mathbf{x}_{1:T} \mid \mathbf{y}_{1:T})} \left[ \nabla_\phi \log q_\phi(\mathbf{x}_{1:T} \mid \mathbf{y}_{1:T}) \right], \quad (4)$$

as derived in Appendix 8.2. This allows RWS to estimate Eq. (4) using SNIS with the same set of samples and weights as Eq. (3),

$$-\mathbb{E}_{p_\theta(\mathbf{x}_{1:T} \mid \mathbf{y}_{1:T})} \left[ \nabla_\phi \log q_\phi(\mathbf{x}_{1:T} \mid \mathbf{y}_{1:T}) \right] \approx -\sum_{i=1}^{N} \overline{w}^{(i)} \nabla_\phi \log q_\phi(\mathbf{x}_{1:T}^{(i)} \mid \mathbf{y}_{1:T}). \quad (5)$$

Importantly, any method that provides estimates of expectations w.r.t. the posterior can be used for gradient estimation within the RWS framework, as we will see in the next section.

## 2.2 Estimating Posterior Expectations with Smoothing Sequential Monte Carlo

As we saw in Eqs. (2) and (4), key quantities in RWS can be expressed as expectations under the posterior. Standard RWS uses SNIS to approximate these expectations, but in sequence models the variance of the SNIS estimator can scale exponentially in the sequence length. In this section, we review sequential Monte Carlo (SMC) [10, 24], an inference algorithm that can produce estimators of posterior expectations with linear or even sub-linear variance scaling.

SMC approximates the posterior $p_\theta(\mathbf{x}_{1:T} \mid \mathbf{y}_{1:T})$ with a set of $N$ weighted particles $\mathbf{x}_{1:T}^{1:N}$ constructed by sampling from a sequence of target distributions $\{\pi_t(\mathbf{x}_{1:t})\}_{t=1}^{T}$. Since these intermediate targets are often only known up to some unknown normalizing constant $Z_t$, SMC uses the unnormalized targets $\{\gamma_t(\mathbf{x}_{1:t})\}_{t=1}^{T}$, where $\pi_t(\mathbf{x}_{1:t}) = \gamma_t(\mathbf{x}_{1:t})/Z_t$. Provided mild technical conditions are met and $\gamma_T(\mathbf{x}_{1:T}) \propto p_\theta(\mathbf{x}_{1:T}, \mathbf{y}_{1:T})$, SMC returns weighted particles that approximate the posterior $p_\theta(\mathbf{x}_{1:T} \mid \mathbf{y}_{1:T})$ [10, 24]. These weighted particles can be used to compute biased but consistent estimates of expectations under the posterior, similar to SNIS.

SMC repeats the following steps for each time $t$:

1. Sample latents $\mathbf{x}_{1:t}^{1:N}$ from a proposal distribution $q_\phi(\mathbf{x}_{1:t} \mid \mathbf{y}_{1:T})$.
2. Weight each particle using the unnormalized target $\gamma_t$ to form an empirical approximation $\hat{\pi}_t$ to the normalized target distribution $\pi_t$.
3. Draw new particles $\mathbf{x}_{1:t}^{1:N}$ from the approximation $\hat{\pi}_t$ (the resampling step).

By resampling away latent trajectories with low weight and focusing on promising particles, SMC can produce lower variance estimates than SNIS. For a thorough review of SMC, see Doucet and Johansen [10], Naesseth et al. [24], and Del Moral [25].

**Filtering vs. Smoothing** The most common choice of unnormalized targets $\gamma_t$ are the *filtering* distributions $p_\theta(\mathbf{x}_{1:t}, \mathbf{y}_{1:t})$, resulting in the algorithm known as filtering SMC or a particle filter. Filtering SMC has been used to estimate posterior expectations within the RWS framework in neural adaptive sequential Monte Carlo (NASMC) [20], but a major disadvantage of filtering SMC is that it ignores future observations $\mathbf{y}_{t+1:T}$. Ignoring future observations can lead to particle degeneracy and high-variance estimates, which in turn causes poor model learning and inference [17, 18, 26, 27].

We could avoid these issues by using the *smoothing* distributions as unnormalized targets, choosing $\gamma_t(\mathbf{x}_{1:t}) = p_\theta(\mathbf{x}_{1:t}, \mathbf{y}_{1:T})$, but unfortunately the smoothing distributions are not readily available from the model. We can approximate them, however, by observing that $p_\theta(\mathbf{x}_{1:t}, \mathbf{y}_{1:T})$ is proportional to the filtering distributions $p_\theta(\mathbf{x}_{1:t}, \mathbf{y}_{1:t})$ times the *lookahead* distributions $p_\theta(\mathbf{y}_{t+1:T} \mid \mathbf{x}_t)$. If the lookahead distributions are well-approximated by a sequence of *twists* $\{r_\psi(\mathbf{y}_{t+1:T}, \mathbf{x}_t)\}_{t=1}^{T}$, then running SMC with targets $\gamma_t(\mathbf{x}_{1:t}) = p_\theta(\mathbf{x}_{1:t}, \mathbf{y}_{1:t}) r_\psi(\mathbf{y}_{t+1:T}, \mathbf{x}_t)$ approximates smoothing SMC [26].

**Twist Learning**   We have reduced the challenge of obtaining the smoothing distributions to learning twists that approximate the lookahead distributions. Previous twist-learning approaches include maximum likelihood training on samples from the model [18, 28] and Bellman-type losses motivated by writing the twist at time $t$ recursively in terms of the twist at time $t+1$ [18, 29]. For NAS-X we use density ratio estimation (DRE) via classification to learn the twists, as introduced in Lawson et al. [19]. This method is motivated by observing that the lookahead distribution is proportional to a ratio of densities up to a constant independent of $x_t$,

$$p_{\boldsymbol{\theta}}(\mathbf{y}_{t+1:T} \mid \mathbf{x}_t) = \frac{p_{\boldsymbol{\theta}}(\mathbf{x}_t \mid \mathbf{y}_{t+1:T})\, p_{\boldsymbol{\theta}}(\mathbf{y}_{t+1:T})}{p_{\boldsymbol{\theta}}(\mathbf{x}_t)} \propto \frac{p_{\boldsymbol{\theta}}(\mathbf{x}_t \mid \mathbf{y}_{t+1:T})}{p_{\boldsymbol{\theta}}(\mathbf{x}_t)}. \tag{6}$$

Results from the DRE via classification literature [30] provide a way to approximate this density ratio: train a classifier to distinguish between samples from the numerator $p_{\boldsymbol{\theta}}(\mathbf{x}_t \mid \mathbf{y}_{t+1:T})$ and denominator $p_{\boldsymbol{\theta}}(\mathbf{x}_t)$. Then, the pre-sigmoid output of the classifier will approximate the log of the ratio in Eq. (6). For an intuitive argument for this fact see Appendix 8.3, and for a full proof see Sugiyama et al. [30].

In practice, it is not possible to sample directly from $p_{\boldsymbol{\theta}}(\mathbf{x}_t \mid \mathbf{y}_{t+1:T})$. Instead, Lawson et al. [19] sample full trajectories from the model's joint distribution, i.e. draw $\mathbf{x}_{1:T}, \mathbf{y}_{1:T} \sim p_{\boldsymbol{\theta}}(\mathbf{x}_{1:T}, \mathbf{y}_{1:T})$, and discard unneeded timesteps, leaving only $\mathbf{x}_t$ and $\mathbf{y}_{t+1:T}$ which are distributed marginally as $p_{\boldsymbol{\theta}}(\mathbf{x}_t, \mathbf{y}_{t+1:T})$. Training the DRE classifier on data sampled in this manner will approximate the ratio $p_{\boldsymbol{\theta}}(\mathbf{x}_t, \mathbf{y}_{t+1:T})/p_{\boldsymbol{\theta}}(\mathbf{x}_t)p_{\boldsymbol{\theta}}(\mathbf{y}_{t+1:T})$, which is equivalent to Eq. (6), see Appendix 8.3.

## 3   NAS-X: Neural Adaptive Smoothing via Twisting

The goal of NAS-X is to combine recent advances in smoothing SMC with the advantages of reweighted wake-sleep. Because SMC is a self-normalized importance sampling algorithm, it can be used to estimate posterior expectations and therefore the model and proposal gradients within a reweighted wake-sleep framework. In particular, NAS-X repeats the following steps:

1. Draw a set of $N$ trajectories $\mathbf{x}_{1:T}^{(1:N)}$ and weights $\overline{w}_{1:T}^{(1:N)}$ from a smoothing SMC run with model $p_\theta$, proposal $q_\phi$, and twist $r_\psi$.

2. Use those trajectories and weights to form estimates of gradients for the model $p_\theta$ and proposal $q_\phi$, as in reweighted wake-sleep. Specifically, NAS-X computes the gradients of the inclusive KL divergence for learning the proposal $q_\phi$ as

$$-\sum_{t=1}^{T} \sum_{i=1}^{N} \overline{w}_t^{(i)} \nabla_\phi \log q_\phi(\mathbf{x}_t^{(i)} \mid \mathbf{x}_{t-1}^{(i)}, \mathbf{y}_{t:T}) \tag{7}$$

   and computes the gradients of the model $p_\theta$ as

$$\sum_{t=1}^{T} \sum_{i=1}^{N} \overline{w}_t^{(i)} \nabla_\theta \log p_\theta(\mathbf{x}_t^{(i)}, \mathbf{y}_t \mid \mathbf{x}_{t-1}^{(i)}). \tag{8}$$

3. Update the twists $r_\psi$ using density ratio estimation via classification.

A full description is available in Algorithms 1 and 2.

A key design decision in NAS-X is the specific form of the gradient estimators. Smoothing SMC provides two ways to estimate expectations of test functions with respect to the posterior: both the timestep-$t$ and timestep-$T$ approximations of the target distribution could be used, in the latter case by discarding timesteps after $t$. Specifically,

$$p_\theta(\mathbf{x}_{1:t} \mid \mathbf{y}_{1:T}) \approx \sum_{i=1}^{N} \overline{w}_t^{(i)} \delta(\mathbf{x}_{1:t}\,;\, \mathbf{x}_{1:t}^{(i)}) \approx \sum_{i=1}^{N} \overline{w}_T^{(i)} \delta(\mathbf{x}_{1:t}\,;\, (\mathbf{x}_{1:T}^{(i)})_{1:t}) \tag{9}$$

where $\delta(a\,;\,b)$ is a Dirac delta of $a$ located at $b$ and $(\mathbf{x}_{1:T})_{1:t}$ denotes selecting the first $t$ timesteps of a timestep-$T$ particle; due to SMC's ancestral resampling step these are not in general equivalent. For NAS-X we choose the time-$t$ approximation of the posterior to lessen particle degeneracy, as in NASMC [20]. Note, however, that in the case of NASMC this amounts to approximating the posterior with the filtering distributions, which ignores information from future observations. In the case of NAS-X, the intermediate distributions directly approximate the posterior distributions because of the twists, a key advantage that we explore theoretically in Section 3.1 and empirically in Section 5.

**Algorithm 1:** NAS-X

**Procedure** NAS-X($\theta_0$, $\phi_0$, $\psi_0$, $\mathbf{y}_{1:T}$)

    $\theta \leftarrow \theta_0$,    $\phi \leftarrow \phi_0$,    $\psi \leftarrow \psi_0$

    **while** *not converged* **do**

        $\mathbf{x}_{1:T}^{1:N}, \overline{w}_{1:T}^{1:N} \leftarrow \text{SMC}(\{p_\theta(\mathbf{x}_{1:t}, \mathbf{y}_{1:t}), q_\phi(\mathbf{x}_t \mid \mathbf{x}_{t-1}, \mathbf{y}_{t:T}), r_\psi(\mathbf{x}_t, \mathbf{y}_{t+1:T})\}_{t=1}^T)$

        $\Delta\theta = \sum_{t=1}^T \sum_{i=1}^N \overline{w}_t^{(i)} \nabla_\theta \log p_\theta(\mathbf{x}_t^{(i)}, \mathbf{y}_t \mid \mathbf{x}_{t-1}^{(i)})$

        $\Delta\phi = -\sum_{t=1}^T \sum_{i=1}^N \overline{w}_t^{(i)} \nabla_\phi \log q_\phi(\mathbf{x}_t^{(i)} \mid \mathbf{x}_{t-1}^{(i)}, \mathbf{y}_{t:T})$

        $\theta \leftarrow \text{grad-step}(\theta, \Delta\theta)$

        $\phi \leftarrow \text{grad-step}(\phi, \Delta\phi)$

        $\psi \leftarrow \text{twist-training}(\theta, \psi)$

    **end**

**return** $\theta, \phi, \psi$

**Procedure** twist-training($\theta$, $\psi_0$)

    See Algorithm 2 in Appendix 8.3.

## 3.1 Theoretical Analysis of NAS-X

In this section, we state two theoretical results that illustrate NAS-X's advantages over NASMC, with proofs given in Appendix 7.

**Proposition 1.** Consistency of NAS-X's gradient estimates. *Suppose the twists are optimal so that $r_\psi(\mathbf{y}_{t+1:T}, \mathbf{x}_t) \propto p(\mathbf{y}_{t+1:T} \mid \mathbf{x}_t)$ up to a constant independent of $\mathbf{x}_t$ for $t = 1, \ldots, T-1$. Let $\hat{\nabla}_\theta \log p_\theta(\mathbf{y}_{1:T})$ be NAS-X's weighted particle approximation to the true gradient of the log marginal likelihood $\nabla_\theta \log p_\theta(\mathbf{y}_{1:T})$. Then $\hat{\nabla}_\theta \log p_\theta(\mathbf{y}_{1:T}) \overset{a.s.}{\to} \nabla_\theta \log p_\theta(\mathbf{y}_{1:T})$ as $N \to \infty$.*

**Proposition 2.** Unbiasedness of NAS-X's gradient estimates. *Assume that proposal distribution $q_\phi(\mathbf{x}_t \mid \mathbf{x}_{1:t-1}, \mathbf{y}_{1:T})$ is optimal so that $q_\phi(\mathbf{x}_t \mid \mathbf{x}_{1:t-1}, \mathbf{y}_{1:T}) = p(\mathbf{x}_t \mid \mathbf{x}_{1:t-1}, \mathbf{y}_{1:T})$ for $t = 1, \ldots, T$, and the twists $r_\psi(\mathbf{y}_{t+1:T}, \mathbf{x}_t)$ are optimal so that $r_\psi(\mathbf{y}_{t+1:T}, \mathbf{x}_t) \propto p(\mathbf{y}_{t+1:T} \mid \mathbf{x}_t)$ up to a constant independent of $\mathbf{x}_t$ for $t = 1, \ldots, T-1$. Let $\hat{\nabla}_\theta \log p_\theta(\mathbf{y}_{1:T})$ be NAS-X's weighted particle approximation to the true gradient of the log marginal likelihood, $\nabla_\theta \log p_\theta(\mathbf{y}_{1:T})$. Then, for any number of particles, $\mathbb{E}[\hat{\nabla}_\theta \log p_\theta(\mathbf{y}_{1:T})] = \log p_\theta(\mathbf{y}_{1:T})$.*

# 4 Related Work

**VI Methods** There is a large literature on model learning via stochastic gradient ascent on an evidence lower bound (ELBO) [31, 32, 4, 33]. Subsequent works have considered ELBOs defined by the normalizing constant estimates from multiple importance sampling [34], nested importance sampling, [35, 36], rejection sampling, and Hamiltonian Monte Carlo [37]. Most relevant to our work is the literature that uses SMC's estimates of the normalizing constant as a surrogate objective. There are a number of VI methods based on filtering [17, 16, 15] and smoothing SMC [18, 38, 39, 19, 27], but filtering SMC approaches can suffer from particle degeneracy and high variance [18, 19].

**Reweighted Wake-Sleep Methods** The wake-sleep algorithm was introduced in Hinton et al. [13] as a way to train deep directed graphical models. Bornschein and Bengio [14] interpreted the wake-sleep algorithm as self-normalized importance sampling and proposed reweighted wake-sleep, which uses SNIS to approximate gradients of the inclusive KL divergence and log marginal likelihood. Neural adaptive sequential Monte Carlo (NASMC) extends RWS by using filtering SMC to approximate posterior expectations instead of SNIS [20]. To combat particle degeneracy, NASMC approximates the posterior with the filtering distributions, which introduces bias.

**NAS-X vs. SIXO** Both NAS-X and SIXO [19] leverage smoothing SMC with DRE-learned twists, but NAS-X uses smoothing SMC to estimate gradients in an RWS-like framework while SIXO uses it within a VI-like framework. Thus, NAS-X follows biased but consistent estimates of the log marginal likelihood while SIXO follows unbiased estimates of a lower bound on the log marginal likelihood. It is not clear a-priori which approach would perform better, but we provide empirical evidence in Section 5 that shows NAS-X is more stable than SIXO and learns better models and proposals. In

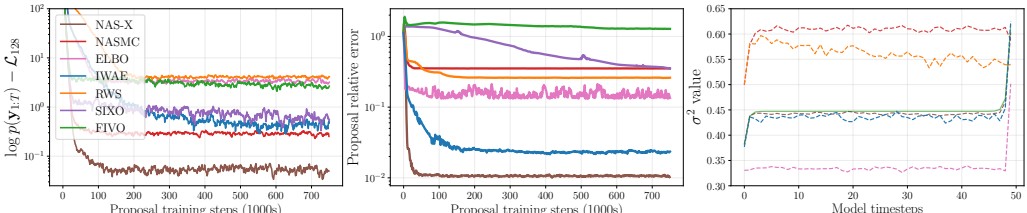

Figure 1: **Comparison of NAS-X and baseline methods on inference in LG-SSM. (left)** Comparison of log-marginal likelihood bounds (lower is better), **(middle)** proposal parameter error (lower is better), and **(right)** learned proposal variances. NAS-X outperforms all baseline methods and recovers the true posterior marginals.

addition to these empirical advantages, NAS-X can fit discrete latent variable models while SIXO would require high-variance score function estimates.

## 5 Experiments

We empirically validate the following advantages of NAS-X:

- By using the approximate smoothing distributions as targets for proposal learning, NAS-X can learn proposals that match the true posterior marginals, while NASMC and other baseline methods cannot. We illustrate this in Section 5.1, in a setting where the true posterior is tractable. We illustrate the practical benefits on inference in nonlinear mechanistic models in Section 5.3.

- By optimizing the proposal within the RWS framework (*e.g.,* descending the inclusive KL), NAS-X can perform learning and inference in discrete latent variable models, which SIXO cannot. We explore this in Section 5.2.

- We explore the practical benefits of this combination in a challenging setting in Section 5.3, where we show NAS-X can fit ODE-based mechanistic models of neural dynamics with 38 model parameters and an 18-dimensional latent state.

In addition to these experiments, we analyze the computational complexity and wall-clock speed of each method and the bias and variance of the gradient estimates in Sections 15 and 16 of the Appendix.

### 5.1 Linear Gaussian State Space Model

We first consider a one-dimensional linear Gaussian state space model with joint distribution

$$p(\mathbf{x}_{1:T}, \mathbf{y}_{1:T}) = \mathcal{N}(\mathbf{x}_1; 0, \sigma_x^2) \prod_{t=2}^{T} \mathcal{N}(\mathbf{x}_{t+1}; \mathbf{x}_t, \sigma_x^2) \prod_{t=1}^{T} \mathcal{N}(\mathbf{y}_t; \mathbf{x}_t, \sigma_y^2). \tag{10}$$

In Figure 1, we compare NAS-X against several baselines (NASMC, FIVO, SIXO, RWS, IWAE, and ELBO) by evaluating log marginal likelihood estimates (left panel) and recovery of the true posterior (middle and right panels). For all methods we learn a mean-field Gaussian proposal factored over time, $q(\mathbf{x}_{1:T}) = \prod_{t=1}^{T} q_t(\mathbf{x}_t) = \prod_{t=1}^{T} \mathcal{N}(\mathbf{x}_t; \mu_t, \sigma_t^2)$, with parameters $\mu_{1:T}$ and $\sigma_{1:T}^2$ corresponding to the means and variances at each time-step. For twist-based methods, we parameterize the twist as a quadratic function in $\mathbf{x}_t$ whose coefficients are functions of the observations and time step. We chose this form to match the functional form of the analytic log density ratio. For details, see Section 9 in the Appendix. NAS-X outperforms all baseline methods, achieving a tighter lower bound on the log-marginal likelihood and lower parameter error.

In the right panel of Figure 1, we compare the learned proposal variances against the true posterior variance, which can be computed in closed form. See Section 9 for comparison of proposal means; we do not report this comparison in the main text since all methods recover the posterior mean. This comparison gives insight into NAS-X's better performance. NASMC's learned proposal overestimates the posterior variance and fails to capture the true posterior distribution, because it employs a filtering

approximation to the gradients of the proposal distribution. On the other hand, by using the twisted targets, which approximate the smoothing distributions, to estimate proposal gradients, NAS-X recovers the true posterior.

## 5.2 Switching Linear Dynamical Systems

To explore NAS-X's ability to handle discrete latent variables, we consider a switching linear dynamical system (SLDS) model [40, 41]. Specifically, we adapt the recurrent SLDS example from Linderman et al. [41] in which the latent dynamics trace ovals in a manner that resembles cars racing on a NASCAR track. There are two coupled sets of latent variables: a discrete state $\mathbf{z}_t$, with $K = 4$ possible values, and a two-dimensional continuous state $\mathbf{x}_t$ that follows linear dynamics that depend on $\mathbf{z}_t$. The observations are a noisy projection of $\mathbf{x}_t$ into a ten-dimensional observation space. There are 1000 observations in total. For the proposal, we factor $q$ over both time and the continuous and discrete states. The continuous distributions are parameterized by Gaussians, and categorical distributions are used for the discrete latent variables. For additional details on the proposal and twist, see Section 10 in the Appendix and for details on the generative model see Linderman et al. [41].

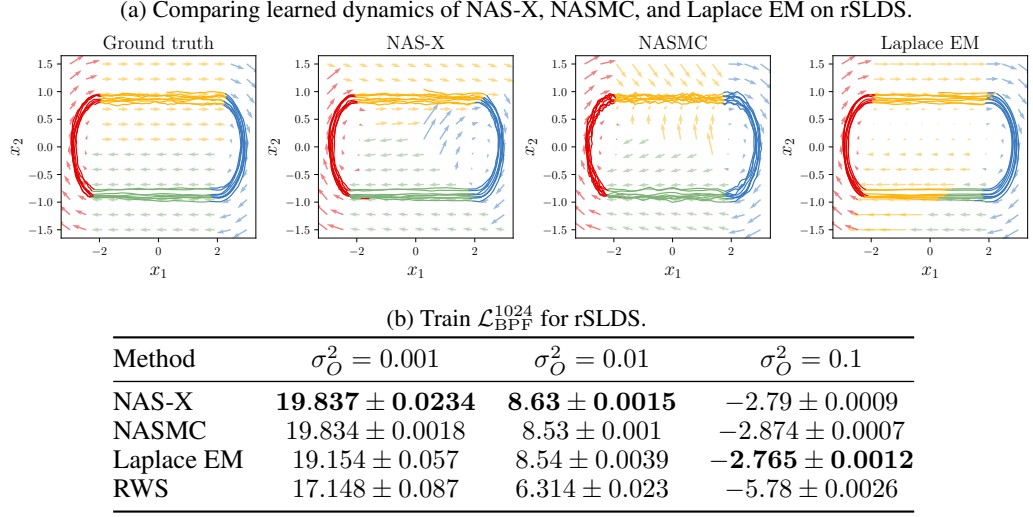

(a) Comparing learned dynamics of NAS-X, NASMC, and Laplace EM on rSLDS.

(b) Train $\mathcal{L}_{\text{BPF}}^{1024}$ for rSLDS.

| Method | $\sigma_O^2 = 0.001$ | $\sigma_O^2 = 0.01$ | $\sigma_O^2 = 0.1$ |
|---|---|---|---|
| NAS-X | $\mathbf{19.837 \pm 0.0234}$ | $\mathbf{8.63 \pm 0.0015}$ | $-2.79 \pm 0.0009$ |
| NASMC | $19.834 \pm 0.0018$ | $8.53 \pm 0.001$ | $-2.874 \pm 0.0007$ |
| Laplace EM | $19.154 \pm 0.057$ | $8.54 \pm 0.0039$ | $\mathbf{-2.765 \pm 0.0012}$ |
| RWS | $17.148 \pm 0.087$ | $6.314 \pm 0.023$ | $-5.78 \pm 0.0026$ |

Figure 2: **Inference and model learning in switching linear dynamical systems (SLDS)**. (**top**) Comparison of learned dynamics and inferred latent states in model learning. Laplace EM sometimes learns incorrect segmentations, as seen in the rightmost panel. (**bottom**) Quantitative comparison of log marginal likelihood lower bounds obtained from running bootstrap particle filter (BPF) with learned models.

We present qualitative results from model learning and inference in the top panel of Figure 2. We compare the learned dynamics for NAS-X, NASMC, and a Laplace-EM algorithm designed specifically for recurrent state space models [42]. In each panel, we plot the vector field of the learned dynamics and the posterior means, with different colors corresponding to the four discrete states. NAS-X recovers the true dynamics accurately. In the Table in Figure 2, we quantitatively compare the model learning performances across these three approaches by running a bootstrap proposal with the learned models and the true dynamics and observation variances. We normalize the bounds by the sequence length. NAS-X outperforms or performs on par with both NASMC and Laplace EM across the different observation noises $\sigma_O^2$. See Section 10, for additional results on inference.

## 5.3 Biophysical Models of Neuronal Dynamics

For our final set of experiments we consider inference and parameter learning in Hodgkin-Huxley (HH) models [9, 43] — mechanistic models of voltage dynamics in neurons. These models use systems of coupled nonlinear differential equations to describe the evolution of the voltage difference across a neuronal membrane as it changes in response to external stimuli such as injected current. Un-

derstanding how voltage propagates throughout a cell is central to understanding electrical signaling and computation in the brain.

Voltage dynamics are governed by the flow of charged ions across the cell membrane, which is in turn mediated by the opening and closing of ion channels and pumps. HH models capture the states of these ion channels as well as the concentrations of ions and the overall voltage, resulting in a complex dynamical system with many free parameters and a high dimensional latent state space. Model learning and inference in this setting can be extremely challenging due to the dimensionality, noisy data, and expensive and brittle simulators that fail for many parameter settings.

**Model Description**   We give a brief introduction to the models used in this section and defer a full description to the appendix. We are concerned with modeling the potential difference across a neuronal cell membrane, $v$, which changes in response to currents flowing through a set of ion channels, $c \in \mathcal{C}$. Each ion channel $c$ has an *activation* which represents a percentage of the maximum current that can flow through the channel and is computed as a nonlinear function $g_c$ of the channel state $\lambda_c$, with $g_c(\lambda_c) \in [0, 1]$. This activation specifies the time-varying conductance of the channel as a fraction of the maximum conductance of the channel, $\overline{g}_c$. Altogether, the dynamics for the voltage $v$ can be written as

$$c_m \frac{dv}{dt} = \frac{I_{\text{ext}}}{S} - \sum_{c \in \mathcal{C}} \overline{g}_c g_c(\lambda_c)(v - E_{c_{\text{ion}}}) \tag{11}$$

where $c_m$ is the specific membrane capacitance, $I_{\text{ext}}$ is the external current applied to the cell, $S$ is the cell membrane surface area, $c_{\text{ion}}$ is the ion transported by channel $c$, and $E_{c_{\text{ion}}}$ is that ion's reversal potential. In addition to the voltage dynamics, the ion channel states $\{\lambda_c\}_{c \in \mathcal{C}}$ evolve as

$$\frac{d\lambda_c}{dt} = A(v)\lambda_c + b(v) \quad \forall c \in \mathcal{C} \tag{12}$$

where $A(v)$ and $b(v)$ are nonlinear functions of the membrane potential that produce matrices and vectors, respectively. Together, equations (11) and (12) define a conditionally linear system of first-order ordinary differential equations (ODEs), meaning that the voltage dynamics are linear if the channel states are known and vice-versa.

Following Lawson et al. [19], we augment the deterministic dynamics with zero-mean, additive Gaussian noise to the voltage $v$ and unconstrained gate states $\text{logit}(\lambda_c)$ at each integration time-step. The observations are produced by adding Gaussian noise to the instantaneous membrane potential.

**Proposal and Twist Parameterization**   For all models in this section we amortize proposal and twist learning across datapoints. SIXO and NAS-X proposals used bidirectional recurrent neural networks (RNNs) [44, 45] with a hidden size of 64 units to process the raw observations and external current stimuli, and then fed the processed observations, previous latent state, and a transformer positional encoding [46] into a 64-unit single-layer MLP that produced the parameters of an isotropic Gaussian distribution over the current latent state. Twists were similarly parameterized with an RNN run in reverse across observations, combined with an MLP that accepts the RNN outputs and latent state and produces the twist values.

**Integration via Strang Splitting**   The HH ODEs are *stiff*, meaning they are challenging to integrate at large step sizes because their state can change rapidly. While small step sizes can ensure numerical stability, they also make methods prohibitively slow. For example, many voltage dynamics of interest unfold over hundreds of milliseconds, which could take upwards of 40,000 integration steps at the standard 0.005 milliseconds per step. Because running our models for even 10,000 steps would be too costly, we developed new numerical integration techniques based on an explicit Strang splitting approach that allowed us to stably integrate at 0.1 milliseconds per step, a 20-time speedup [47]. For details, see Section 13 in the Appendix.

### 5.3.1   Hodgkin-Huxley Inference Results

First, we evaluated NAS-X, NASMC, and SIXO in their ability to infer underlying voltages and channel states from noisy voltage observations. For this task we sampled 10,000 noisy voltage traces from a probabilistic Hodgkin-Huxley model of the squid giant axon [9], and used each method to

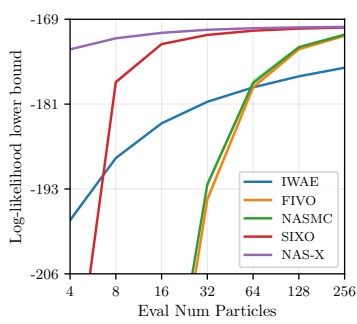

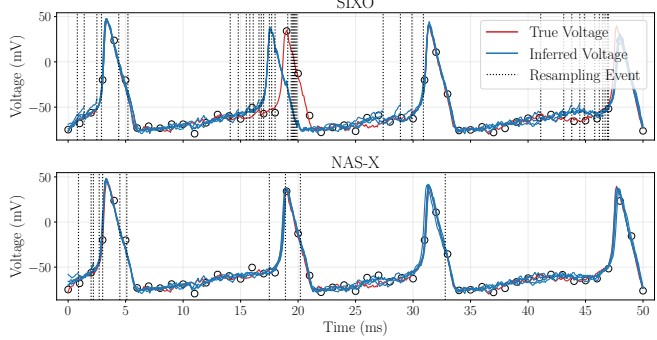

(a) **HH inference performance.**
Proposals were trained with 4 particles and evaluated across a range of particle numbers. RWS performed too poorly to be included.

(b) **Inferred voltage traces for SIXO and NAS-X.**
**(top)** SIXO generates a high number of resampling events leading to particle degeneracy and a single mistimed spike. **(bottom)** NAS-X perfectly infers the latent voltage with no mistimed spikes, and resamples infrequently. See Fig. 9 in Appendix for NASMC's traces.

Figure 3: **Inference in Mechanistic HH Model**

train proposals (and twists for NAS-X and SIXO) to compute the marginal likelihood assigned to the data under the true model. As in [19], we sampled trajectories of length 50 milliseconds, with a single noisy voltage observation every millisecond. The stability of the Strang splitting based ODE integrator allowed us to integrate at $dt = 0.1$ms, meaning there were 10 latent states per observation.

In Figure (3a) we plot the performance of proposals and twists trained with 4 particles and evaluated across a range of particle numbers. All methods perform roughly the same when evaluated with 256 particles, but with lower numbers of evaluation particles the smoothing methods emerge as more particle-efficient than the filtering methods. To achieve NAS-X's inference performance with 4 particles, NASMC would need 256 particles, a 64x increase, and NAS-X is also on average 2x more particle-efficient than SIXO.

In Figure 3b we further investigate these results by examining the inferred voltage traces of NAS-X and SIXO. SIXO accurately infers the timing of most spikes but resamples at a high rate, which can lead to particle degeneracy and poor bound performance. NAS-X correctly infers the voltage across the whole trace with no spurious or mistimed spikes and almost no resampling events, indicating it has learned a high-quality proposal that does not generate poor particles that must be resampled away. These qualitative results support the quantitative results in Figure 3a: SIXO's high resampling rate and NASMC's filtering approach lead to lower bound values.

These results highlight a benefit of RWS-based methods over VI methods: when the model is correctly specified, it can be beneficial to have a more deterministic proposal. Empirically, we find that maximizing the variational lower bound encourages the proposal to have high entropy, which in this case resulted in SIXO's poorer performance relative to NAS-X. In the next section, we explore the implications of this on model learning.

### 5.3.2 Hodgkin-Huxley Model Learning Results

In this section, we assess NAS-X and SIXO's ability to fit model parameters in a more complex, biophysically realistic model of a pyramidal neuron from the mouse visual cortex. This model was taken from the Allen Institute Brain Atlas [48] and includes 9 different voltage-gated ion channels as well as a calcium pump/buffer subsystem and a calcium-gated potassium ion channel. In total, the model had 38 free parameters and an 18-dimensional latent state space, in contrast to the 1 free parameter and 4-dimensional state space of the model considered by Lawson et al. [19]. For full details of the models, see Appendix Section 12.

We fit these models to voltage traces gathered from a real mouse neuron by the Allen Institute, but downsampled and noised the data to simulate a more common voltage imaging setting. We ran a hyperparameter sweep over learning rates and initial values of the voltage and observation noise variances (270 hyperparameter settings in all), and selected the best performing model via early stopping on the train log marginal likelihood lower bound. Each hyperparameter setting was run for

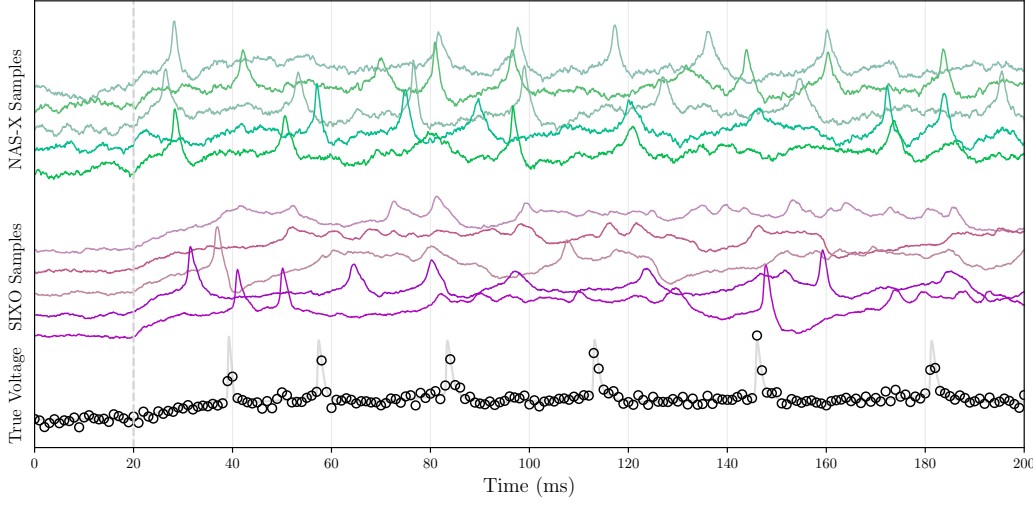

| Method | $\mathcal{L}_{\mathrm{BPF}}^{32}$ | # Spikes Err. | Rest Voltage Err. | Cross-Corr. | % Runs Failed |
|---|---|---|---|---|---|
| NAS-X | $-686.4 \pm 6.8$ | $\mathbf{0.76 \pm 0.15}$ | $2.74 \pm 0.1$ | $\mathbf{6258 \pm 11}$ | $\mathbf{18.9}$ |
| SIXO | $\mathbf{-660.6 \pm 4.4}$ | $1.88 \pm 0.41$ | $\mathbf{1.8 \pm 0.2}$ | $6055 \pm 22$ | $25.2$ |

Figure 4: **Model learning in HH model of a mouse pyramidal neuron (top)** Samples drawn from learned models when stimulated with a square pulse of 250 picoamps beginning at 20 milliseconds (vertical grey dashed line). NAS-X's samples are noisier than SIXO's, but spike more consistently. **(bottom)** A comparison of NAS-X- and SIXO-trained models along various evaluation metrics. SIXO's models achieve higher bounds, but are less stable and capture overall spike count more poorly than NAS-X-trained models. All errors are absolute errors.

5 seeds, and each seed was run for 2 days on a single CPU core with 7 Gb of memory. Because of the inherent instability of these models, many seeds failed, and we discarded hyperparameter settings with more than 2 failed runs.

In Figure 4 (bottom), we compare NAS-X and SIXO-trained models with respect to test set log-likelihood lower bounds as well as biophysically relevant metrics. To compute the biophsyical metrics, we sampled 32 voltage traces for each input stimulus trace in the test set, and averaged the feature errors over the samples and test set. NAS-X better captures the number of spikes, an important feature of the traces, and attains a higher cross correlation. Both methods capture the resting voltage well, although SIXO attains a slightly lower error and outperforms NAS-X in terms of log-likelihood lower bound.

Training instability is a significant practical challenge when fitting mechanistic models. Therefore, we also include the percentage of runs that failed for each method. SIXO's more entropic proposals more frequently generate biophysically implausible latent states, causing the ODE integrator to return NaNs. In contrast, fewer of NAS-X's runs suffer from numerical instability issues, a great advantage when working with mechanistic models.

# 6 Conclusion

In this work we presented NAS-X, a new method for model learning and inference in sequential latent variable models, that combines reweighted wake-sleep framework and approximate smoothing SMC. Our approach involves learning twist functions to use in smoothing SMC, and then running smoothing SMC to approximate gradients of the log marginal likelihood with respect to the model parameters and gradients of the inclusive KL divergence with respect to the proposal parameters. We validated our approach in experiments including model learning and inference for discrete latent variable models and mechanistic models of neural dynamics, demonstrating that NAS-X offers compelling advantages in many settings.

## Acknowledgements

This work was supported by the Simons Collaboration on the Global Brain (SCGB 697092), NIH (U19NS113201, R01NS113119, and R01NS130789), NSF (2223827), Sloan Foundation, McKnight Foundation, the Stanford Center for Human and Artificial Intelligence, and the Stanford Data Science Institute.

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

# 7 Theoretical Analyses

**Proposition 1.** Consistency of NAS-X's gradient estimates. *Suppose the twists are optimal so that $r_\psi(\mathbf{y}_{t+1:T}, \mathbf{x}_t) \propto p(\mathbf{y}_{t+1:T} \mid \mathbf{x}_t)$ up to a constant independent of $\mathbf{x}_t$ for $t = 1, \ldots, T - 1$. Let $\hat{\nabla}_\theta \log p_\theta(\mathbf{y}_{1:T})$ be NAS-X's weighted particle approximation to the true gradient of the log marginal likelihood $\nabla_\theta \log p_\theta(\mathbf{y}_{1:T})$. Then $\hat{\nabla}_\theta \log p_\theta(\mathbf{y}_{1:T}) \overset{a.s.}{\to} \nabla_\theta \log p_\theta(\mathbf{y}_{1:T})$ as $N \to \infty$.*

*Proof.* This is a direct application of Theorem 7.4.3 from Del Moral [25]. SMC methods provide strongly consistent estimates of expectations of test functions with respect to a normalized target distribution. That is, consider some test function $h$ and let SMC's particle weights be denoted as $w_i^t$. We have that $\sum_{i=1}^{K} w_t^k h(\mathbf{x}_{1:t}^k) \overset{a.s.}{\to} \int h(\mathbf{x}_{1:t}) \pi_t(\mathbf{x}_{1:t})$ as $K \to \infty$ where $\pi_t(\mathbf{x}_{1:t})$ is the normalized target distribution. NAS-X sets $\pi_t(\mathbf{x}_{1:t}) \propto \gamma_t(\mathbf{x}_{1:t}) = p_\theta(\mathbf{x}_{1:t}, \mathbf{y}_{1:t}) r_{\psi^*}(\mathbf{y}_{t+1:T}, \mathbf{x}_t)$, which by assumption is proportional to $p_\theta(\mathbf{x}_{1:t}, \mathbf{y}_{1:t}) p_\theta(\mathbf{y}_{t+1:T} \mid \mathbf{x}_t) = p_\theta(\mathbf{x}_{1:t}, \mathbf{y}_{1:T})$. The desired result follows immediately. □

**Proposition 2.** Unbiasedness of NAS-X's gradient estimates. *Assume that proposal distribution $q_\phi(\mathbf{x}_t \mid \mathbf{x}_{1:t-1}, \mathbf{y}_{1:T})$ is optimal so that $q_\phi(\mathbf{x}_t \mid \mathbf{x}_{1:t-1}, \mathbf{y}_{1:T}) = p(\mathbf{x}_t \mid \mathbf{x}_{1:t-1}, \mathbf{y}_{1:T})$ for $t = 1, \ldots, T$, and the twists $r_\psi(\mathbf{y}_{t+1:T}, \mathbf{x}_t)$ are optimal so that $r_\psi(\mathbf{y}_{t+1:T}, \mathbf{x}_t) \propto p(\mathbf{y}_{t+1:T} \mid \mathbf{x}_t)$ up to a constant independent of $\mathbf{x}_t$ for $t = 1, \ldots, T - 1$. Let $\hat{\nabla}_\theta \log p_\theta(\mathbf{y}_{1:T})$ be NAS-X's weighted particle approximation to the true gradient of the log marginal likelihood, $\nabla_\theta \log p_\theta(\mathbf{y}_{1:T})$. Then, for any number of particles, $\mathbb{E}[\hat{\nabla}_\theta \log p_\theta(\mathbf{y}_{1:T})] = \log p_\theta(\mathbf{y}_{1:T})$.*

*Proof.* We first provide a proof sketch then give a more detailed derivation, adapting the proof of a similar theorem in Lawson et al. [19]. We will prove that particles produced from running SMC with the smoothing targets and optimal proposal are exact samples from the posterior distribution of interest. The claim then follows immediately. Under the stated assumptions, both NAS-X and NASMC propose particles from the true posterior. However, the different intermediate target distributions will affect how these particles are distributed after reweighting. For NAS-X, the particles will have equal weight since they are reweighted using the smoothing targets. Thus, after reweighting, the particles are still samples from the true posterior. In contrast, in NASMC, the samples are reweighted by filtering targets and will not be distributed as samples from the true posterior.

We first consider NAS-X which uses the smoothing targets. We will show that the particles drawn at each timestep are sampled from the posterior distribution. This will follow from the fact that the particle weights at each timestep in the SMC sweep are equal; that is, $w_t^k = 1$ or $w_t^k = p(\mathbf{y}_{1:T})$ for $k = 1, \ldots, K$, depending on whether resampling occurred. We proceed by induction on $t$, the timestep in the SMC sweep.

For $t = 1$ note that

1. $\gamma_1(\mathbf{x}_1^k) = p(\mathbf{x}_1^k, \mathbf{y}_1) p(\mathbf{y}_{2:T} \mid \mathbf{x}_1^k)$,

2. $\gamma_0 \triangleq 1$,

3. and $q_1(\mathbf{x}_1^k) = p(\mathbf{x}_1^k \mid \mathbf{y}_{1:T})$.

This implies that the incremental weight $\alpha_1^k$ is

$$\alpha_1^k = \frac{p(\mathbf{x}_1^k, \mathbf{y}_1) p(\mathbf{y}_{2:T} \mid \mathbf{x}_1^k)}{p(\mathbf{x}_1^k \mid \mathbf{y}_{1:T})} = \frac{p(\mathbf{x}_1^k, \mathbf{y}_{1:T})}{p(\mathbf{x}_1^k \mid \mathbf{y}_{1:T})} = p(\mathbf{y}_{1:T}) \tag{13}$$

which does not depend on $k$. Because $w_0^k \triangleq 1$, we have that $w_1^k = w_0^k \alpha_1^k = p(\mathbf{y}_{1:T})$ for all $k$.

Note that, since the proposal is optimal, prior to SMC's reweighting step, the particles were distributed as follows $\mathbf{x}_1^k \sim p(\mathbf{x}_1 \mid \mathbf{y}_{1:T})$. Since the incremental weights are equal, the distribution of particles is unchanged.

For the induction step, assume that $w_{t-1}^{1:K}$ equals 1 or $p(\mathbf{y}_{1:T})$ and that $\mathbf{x}_{1:t-1}^k \sim p(\mathbf{x}_{1:t-1} \mid \mathbf{y}_{1:T})$. The particles are distributed as follows, $\mathbf{x}_t^k \sim p(\mathbf{x}_t \mid \mathbf{x}_{1:t-1}, \mathbf{y}_{1:T})$. This implies that $(\mathbf{x}_{1:t-1}^k, \mathbf{x}_t^k) \sim p(\mathbf{x}_t \mid \mathbf{x}_{1:t-1}, \mathbf{y}_{1:T}) p(\mathbf{x}_{1:t-1} \mid \mathbf{y}_{1:T}) = p(\mathbf{x}_{1:t} \mid \mathbf{y}_{1:T})$.

We now show that the incremental particle weights $\alpha_t^k$ are equal using the following identities/assumptions

1. $\gamma_t(\mathbf{x}_{1:t}^k) = p(\mathbf{x}_{1:t}^k, \mathbf{y}_{1:t})p(\mathbf{y}_{t+1:T} \mid \mathbf{x}_t^k)$,

2. $\gamma_{t-1}(\mathbf{x}_{1:t-1}^k) = p(\mathbf{x}_{1:t-1}^k, \mathbf{y}_{1:t-1})p(\mathbf{y}_{t:T} \mid \mathbf{x}_{t-1}^k)$,

3. $r_\psi(\mathbf{y}_{t+1:T}, \mathbf{x}_t) \propto p(\mathbf{y}_{t+1:T} \mid \mathbf{x}_t)$ up to a constant independent of $\mathbf{x}_t$ for $t = 1, \ldots, T-1$.

4. and $q_t(\mathbf{x}_t^k) = p(\mathbf{x}_t^k \mid \mathbf{x}_{1:t-1}^k, \mathbf{y}_{1:T})$

This shows that $\alpha_t^k$ is given by

$$\alpha_t^k = \frac{p(\mathbf{x}_{1:t}^k, \mathbf{y}_{1:t})p(\mathbf{y}_{t+1:T} \mid \mathbf{x}_t^k)}{p(\mathbf{x}_{1:t-1}^k, \mathbf{y}_{1:t-1})p(\mathbf{y}_{t:T} \mid \mathbf{x}_{t-1}^k)p(\mathbf{x}_t^k \mid \mathbf{x}_{1:t-1}^k, \mathbf{y}_{1:T})} \tag{14}$$

$$= \frac{p(\mathbf{x}_{1:t}^k, \mathbf{y}_{1:T})}{p(\mathbf{x}_{1:t-1}^k, \mathbf{y}_{1:T})p(\mathbf{x}_t^k \mid \mathbf{x}_{1:t-1}^k, \mathbf{y}_{1:T})} \tag{15}$$

$$= \frac{p(\mathbf{x}_{1:t-1}^k, \mathbf{y}_{1:T})p(\mathbf{x}_t^k \mid \mathbf{x}_{1:t-1}, \mathbf{y}_{1:T})}{p(\mathbf{x}_{1:t-1}^k, \mathbf{y}_{1:T})p(\mathbf{x}_t^k \mid \mathbf{x}_{1:t-1}^k, \mathbf{y}_{1:T})} \tag{16}$$

$$= 1 \tag{17}$$

for $k = 1, \ldots, K$.

There are two cases depending on the value of the weights at the previous timestep. If $w_{t-1}^{1:K} = 1$, then $w_t^k = w_{t-1}^k \alpha_t^k = 1$ for all $k$. On the other hand, if $w_{t-1}^{1:K} = p(\mathbf{y}_{1:T})$ then $w_t^k = p(\mathbf{y}_{1:T})$ for all $k$. Therefore, even after reweighting, the particles are still drawn from the true posterior distribution. If resampling occurs, since the weights are equal in both cases, the distribution of the particles remains unchanged.

To conclude, we note that the incremental particle weights are, in general, not the same for NASMC. To see this, consider NASMC's incremental weights at timestep 1.

$$\alpha_1^k = \frac{p(\mathbf{x}_1^k, \mathbf{y}_1)}{p(\mathbf{x}_t^k \mid \mathbf{y}_{1:T})} \tag{18}$$

After reweighting, the particles will be distributed according to the filtering distributions. The distribution of particles after reweighting is proportional to $p(\mathbf{x}_1^k \mid \mathbf{y}_1)$. This is because the distribution of the reweighted particles is proportional to the incremental weights times the optimal proposal distribution. Therefore, the term in the denominator corresponding to the proposal cancels out with the proposal distribution term.

Interestingly, the particle weights will be the same at each iteration under a certain dependency structure for the model $p(\mathbf{x}_{1:t-1} \mid \mathbf{y}_{1:t}) = p(\mathbf{x}_{1:t-1} \mid \mathbf{y}_{1:t-1})$ that was identified in Maddison et al. [17]. However, this dependency is not satisfied in general and therefore NASMC's gradients are not unbiased. $\qquad\square$

# 8 Derivations

## 8.1 Gradient of the Marginal Likelihood

We derive the gradients for the marginal likelihood. This identity is known as Fisher's identity.

$$\nabla_\theta \log p(\mathbf{y}_{1:T}) = \nabla_\theta \log \int p_\theta(\mathbf{x}_{1:T}, \mathbf{y}_{1:T}) d\mathbf{y}_{1:T}$$

$$= \frac{1}{p_\theta(\mathbf{y}_{1:T})} \nabla_\theta \int p_\theta(\mathbf{x}_{1:T}, \mathbf{y}_{1:T}) d\mathbf{x}_{1:T}$$

$$= \frac{1}{p_\theta(\mathbf{y}_{1:T})} \int \nabla_\theta p_\theta(\mathbf{x}_{1:T}, \mathbf{y}_{1:T}) d\mathbf{x}_{1:T}$$

$$= \frac{1}{p_\theta(\mathbf{y}_{1:T})} \int p_\theta(\mathbf{x}_{1:T}, \mathbf{y}_{1:T}) \nabla_\theta \log p_\theta(\mathbf{x}_{1:T}, \mathbf{y}_{1:T}) d\mathbf{x}_{1:T}$$

$$= \int p_\theta(\mathbf{x}_{1:T}|\mathbf{y}_{1:T}) \nabla_\theta \log p_\theta(\mathbf{x}_{1:T}, \mathbf{y}_{1:T}) d\mathbf{x}_{1:T}$$

$$= \int p_\theta(\mathbf{x}_{1:T}|\mathbf{y}_{1:T}) \nabla_\theta \sum_t \log p_\theta(\mathbf{y}_t, \mathbf{x}_t|\mathbf{x}_{t-1}) d\mathbf{x}_{1:T}$$

$$= \sum_t \int p_\theta(\mathbf{x}_{1:T}|\mathbf{y}_{1:T}) \nabla_\theta \log p_\theta(\mathbf{y}_t, \mathbf{x}_t|\mathbf{x}_{t-1}) d\mathbf{x}_{1:T}$$

$$= \sum_t \mathbb{E}_{p_\theta(\mathbf{x}_{1:T}|\mathbf{y}_{1:T})} \left[ \nabla_\theta \log p_\theta(\mathbf{y}_t, \mathbf{x}_t|\mathbf{x}_{t-1}) \right]$$

The key steps were the log-derivative trick and Bayes rule.

## 8.2 Gradient of Inclusive KL Divergence

Below, we derive the gradient of the inclusive KL divergence for a generic Markovian model. In this derivation, we assume there are no shared parameters between the proposal and model.

$$-\nabla_\phi \mathrm{KL}(p_\theta||q_\phi) = \nabla_\phi \int p_\theta(\mathbf{x}_{1:T}|\mathbf{y}_{1:T}) \log q_\phi(\mathbf{x}_{1:T}|\mathbf{y}_{1:T}) d\mathbf{x}_{1:T}$$

$$= \int p_\theta(\mathbf{x}_{1:T}|\mathbf{y}_{1:T}) \nabla_\phi \log q_\phi(\mathbf{x}_{1:T}|\mathbf{y}_{1:T}) d\mathbf{x}_{1:T}$$

$$= \int p_\theta(\mathbf{x}_{1:T}|\mathbf{y}_{1:T}) \nabla_\phi \left( \sum_t \log q_\phi(\mathbf{x}_t|\mathbf{x}_{t-1}, \mathbf{y}_{t:T}) \right) d\mathbf{x}_{1:T}$$

$$= \sum_t \int p_\theta(\mathbf{x}_{1:T}|\mathbf{y}_{1:T}) \nabla_\phi \log q_\phi(\mathbf{x}_t|\mathbf{x}_{t-1}, \mathbf{y}_{t:T}) d\mathbf{x}_{1:T}$$

$$= \sum_t \mathbb{E}_{p_\theta(\mathbf{x}_{1:T}|\mathbf{y}_{1:T})} \left[ \nabla_\phi \log q_\phi(\mathbf{x}_t|\mathbf{x}_{t-1}, \mathbf{y}_{t:T}) \right]$$

We use the assumption that there are no shared parameters in the second equality.

## 8.3 Density Ratio Estimation via Classification

Here we briefly summarize density ratio estimation (DRE) via classification. For a full treatment, see Sugiyama et al. [30].

Let $a(x)$ and $b(x)$ be two distributions defined over the same space $\mathcal{X}$, and consider a classifier $g : \mathcal{X} \to \mathbb{R}$ that accepts a specific $x \in \mathcal{X}$ and classifies it as either being sampled from $a$ or $b$. We will train this classifier to predict whether a given $x$ was sampled from $a(x)$ or $b(x)$. The raw outputs (logits) of this classifier will approximate $\log(a(x)/b(x))$ up to a constant that does not depend on $x$.

To see this, define an expanded generative model where we first sample $z \in \{0, 1\}$ from a Bernoulli random variable with probability $0 < \rho < 1$, and then sample $x$ from $a(x)$ if $z = 1$, and sample $x$

from $b(x)$ if $z = 0$. This defines the joint distribution

$$p(x, z) = p(z)p(x|z) = \text{Bernoulli}(z; \rho)a(x)^z b(x)^{(1-z)}, \tag{19}$$

where $p(x \mid z = 1) = a(x)$ and $p(x \mid z = 0) = b(x)$.

Let $g : \mathcal{X} \to \mathbb{R}$ be a function that accepts $x \in \mathcal{X}$ and produces the logit for Bernoulli distribution over $z$. This function will parameterize a classifier via the sigmoid function, meaning that the classifier's Bernoulli conditional distribution over $z$ is defined as

$$p_g(z|x) \triangleq \sigma(g(x))^z (1 - \sigma(g(x)))^{1-z}, \quad z \in \{1, 0\} \tag{20}$$

where $\sigma$ is the sigmoid function and $\sigma^{-1}$ is its inverse, the logit function

$$\sigma(\ell) = \frac{1}{1 + e^{-\ell}}, \quad \sigma^{-1}(p) = \log\left(\frac{p}{1-p}\right). \tag{21}$$

The optimal function $g^*$ will be selected by solving the maximum likelihood problem

$$g^* \triangleq \arg\max_g \mathbb{E}_{p(x,z)} \left[ p_g(z \mid x) \right]. \tag{22}$$

The solution to this problem is the true $p(z \mid x)$. Because we have not restricted $g$, this solution can be obtained. Thus,

$$p(z = 1 \mid x) = p_{g^*}(z = 1 \mid x) \tag{23}$$
$$= \sigma(g^*(x))^1 (1 - \sigma(g^*(x)))^{1-1} \tag{24}$$
$$= \sigma(g^*(x)). \tag{25}$$

This in turn implies that

$$g^*(x) = \sigma^{-1}(p(z = 1 \mid x)) \tag{26}$$
$$= \log\left(\frac{p(z = 1 \mid x)}{1 - p(z = 1 \mid x)}\right) \tag{27}$$
$$= \log\left(\frac{p(z = 1 \mid x)}{p(z = 0 \mid x)}\right) \tag{28}$$
$$= \log\left(\frac{p(z = 1 \mid x)p(x)}{p(z = 0 \mid x)p(x)}\right) \tag{29}$$
$$= \log\left(\frac{p(z = 1, x)}{p(z = 0, x)}\right) \tag{30}$$
$$= \log\left(\frac{p(x \mid z = 1)p(z = 1)}{p(x \mid z = 0)p(z = 0)}\right) \tag{31}$$
$$= \log\left(\frac{p(x \mid z = 1)}{p(x \mid z = 0)}\right) + \log\left(\frac{p(z = 1)}{p(z = 0)}\right) \tag{32}$$
$$= \log\left(\frac{a(x)}{b(x)}\right) + \log\left(\frac{\rho}{1 - \rho}\right) \tag{33}$$

Thus, the optimal solution to the classification problem, $g^*$, is proportional to $\log(a(x)/b(x))$ up to a constant that does not depend on $x$. In practice we observe empirically that as long as a sufficiently flexible parametric family for $g$ is selected, $g^*$ will closely approximate the desired density ratio.

In the case of learning the ratio required for smoothing SMC,

$$\frac{p_{\boldsymbol{\theta}}(\mathbf{x}_t \mid \mathbf{y}_{t+1:T})}{p_{\boldsymbol{\theta}}(\mathbf{x}_t)}, \tag{34}$$

Lawson et al. [19] instead learn the equivalent ratio

$$\frac{p_{\boldsymbol{\theta}}(\mathbf{x}_t, \mathbf{y}_{t+1:T})}{p_{\boldsymbol{\theta}}(\mathbf{x}_t)p_{\boldsymbol{\theta}}(\mathbf{y}_{t+1:T})}. \tag{35}$$

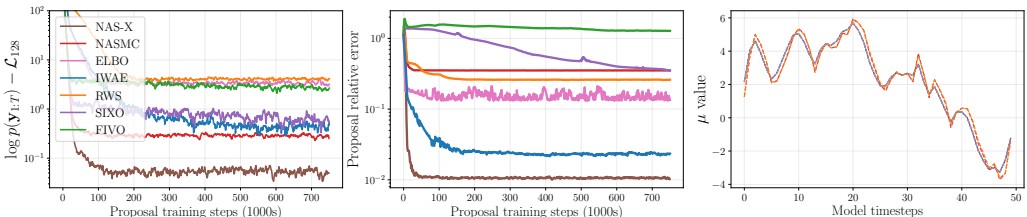

Figure 5: **Comparison of NAS-X vs baseline methods on Inference in LG-SSM.** (**left**) Comparison of log-marginal likelihood bounds (lower is better), (**middle**) proposal parameter error (lower is better), and (**right**) learned proposal means. NAS-X outperforms several baseline methods and recovers the true posterior marginals.

As per the previous derivation, it suffices to train a classifier to distinguish between samples from the numerator and denominator of Eq. 35. To accomplish this, Lawson et al. [19] draw paired and unpaired samples from the model that are distributed marginally according to the desired densities. Specifically, consider drawing

$$\mathbf{x}_{1:T}, \mathbf{y}_{1:T} \sim p_{\boldsymbol{\theta}}(\mathbf{x}_{1:T}, \mathbf{y}_{1:T}) \quad \tilde{\mathbf{x}}_{1:T} \sim p_{\boldsymbol{\theta}}(\mathbf{x}_{1:T}) \tag{36}$$

and note that any $\mathbf{x}_t, \mathbf{y}_{t+1:T}$ selected from the sample will be distributed marginally according to $p_{\boldsymbol{\theta}}(\mathbf{x}_t, \mathbf{y}_{t+1:T})$. Similarly, any $\tilde{\mathbf{x}}_t, \mathbf{y}_{t+1:T}$ will be distributed marginally as $p_{\boldsymbol{\theta}}(\mathbf{x}_t)p_{\boldsymbol{\theta}}(\mathbf{y}_{t+1:T})$. In this way, $T-1$ positive and negative training examples for the DRE classifier can be drawn using a single set of samples as in Eq. (36).

The twist training process is summarized in Algorithm 2.

---

**Algorithm 2:** Twist Training

**Procedure** twist-training($\theta, \psi_0$)
  $\psi \leftarrow \psi_0$
  **while** *not converged* **do**
    $\mathbf{x}_{1:T}, \mathbf{y}_{1:T} \sim p_{\boldsymbol{\theta}}(\mathbf{x}_{1:T}, \mathbf{y}_{1:T})$
    $\tilde{\mathbf{x}}_{1:T} \sim p_{\boldsymbol{\theta}}(\mathbf{x}_{1:T})$
    $\mathcal{L}(\psi) = \frac{1}{T-1} \sum_{t=1}^{T-1} \log \sigma(r_\psi(\mathbf{x}_t, \mathbf{y}_{t+1:T})) + \log(1 - \sigma(r_\psi(\tilde{\mathbf{x}}_t, \mathbf{y}_{t+1:T})))$
    $\psi \leftarrow$ grad-step($\psi, \nabla_\psi \mathcal{L}(\psi)$)
  **end**
**return** $\psi$

---

# 9 LGSSM

**Model Details**    We consider a one-dimensional linear Gaussian state space model with joint distribution

$$p(\mathbf{x}_{1:T}, \mathbf{y}_{1:T}) = \mathcal{N}(\mathbf{x}_1; 0, \sigma_x^2) \prod_{t=2}^{T} \mathcal{N}(\mathbf{x}_{t+1}; \mathbf{x}_t, \sigma_x^2) \prod_{t=1}^{T} \mathcal{N}(\mathbf{y}_t; \mathbf{x}_t, \sigma_y^2). \tag{37}$$

In our experiments we set the dynamics variance $\sigma_x^2 = 1.0$ and the observation variance $\sigma_y^2 = 1.0$.

**Proposal Parameterization**    For both NAS-X and NASMC, we use a mean-field Gaussian proposal factored over time

$$q(x_{1:T}) = \prod_{t=1}^{T} q_t(x_t) = \prod_{t=1}^{T} \mathcal{N}(x_t; \mu_t, \sigma_t^2), \tag{38}$$

with parameters $\mu_{1:T}$ and $\sigma_{1:T}^2$ corresponding to the means and variances at each timestep. In total, we learn $2T$ proposal parameters.

**Twist Parametrization** We parameterize the twist as a quadratic function in $x_t$ whose coefficients are functions of the observations and time step and are learned via the density ratio estimation procedure described in [19]. We chose this form to match the analytic log density ratio for the model defined in Eq 10. Given that $p(x_{1:T}, y_{1:T})$ is a multivariate Gaussian, we know that $p(x_t \mid y_{t+1:T})$ and $p(x_t)$ are both marginally Gaussian. Let

$$p(x_t \mid y_{t+1:T}) \triangleq \mathcal{N}(\mu_1, \sigma_1^2)$$
$$p(x_t) \triangleq \mathcal{N}(0, \sigma_1^2)$$

Then,

$$\log\left(\frac{p(x_t \mid y_{t+1:T})}{p(x_t)}\right) = \log \mathcal{N}(x_t; \mu_1, \sigma_1^2) - \log \mathcal{N}(x_t; 0, \sigma_2^2)$$

$$= \log Z(\sigma_1) - \frac{1}{2\sigma_1^2}x_t^2 + \frac{\mu_1}{\sigma_1^2}x_t - \frac{\mu_1^2}{2\sigma_1^2} - \log Z(\sigma_2) + \frac{1}{2\sigma^2}x_t^2$$

where $Z(\sigma) = \frac{1}{\sigma\sqrt{2\pi}}$, so $\log Z(\sigma) = -\log(\sigma\sqrt{2\pi})$.

Collecting terms gives:

$$-\log(\sigma_1\sqrt{2\pi}) + \log(\sigma_2\sqrt{2\pi})$$
$$-\frac{1}{2}\left(\frac{1}{\sigma_1^2} - \frac{1}{\sigma_2^2}\right)x_t^2$$
$$+\frac{\mu_1}{\sigma_1^2}x_t$$
$$-\frac{\mu_1^2}{2\sigma_1^2}$$

So we'll define

$$a \triangleq -\frac{1}{2}\left(\frac{1}{\sigma_1^2} - \frac{1}{\sigma_2^2}\right)$$
$$b \triangleq \frac{\mu_1}{\sigma_1^2}$$
$$c \triangleq -\frac{\mu_1^2}{2\sigma_1^2} - \log(\sigma_1\sqrt{2\pi}) + \log(\sigma_2\sqrt{2\pi})$$

We'll explicitly model $\log \sigma_1^2$, $\log \sigma_2^2$ and $\mu_1$. Both $\log \sigma_1^2$ and $\log \sigma_2^2$ are only functions of $t$, not of $y_{t+1:T}$, so those can be vectors of shape $T$ initialized at 0. $\mu_1$ is a linear function of $y_{t+1:T}$ and $t$, so that can be parameterized by a set of $T \times T$ weights, initialized to $1/T$ and $T$ biases initialized to 0.

**Training Details** We use a batch size of 32 for the density ratio estimation step. Since we do not perform model learning, we do not repeatedly alternate between twist training and proposal training for NAS-X. Instead, we first train the twist for 3,000,000 iterations with a batch size of 32 using samples from the model. We then train the proposal for $750,000$ iterations. For the twist, we used Adam with a learning rate schedule that starts with a constant learning rate of $1e-3$, decays the learning by 0.3 and 0.33 at $100,000$ and $300,000$ iterations. For the proposal, we used Adam with a constant learning rate of $1e-3$. For NASMC, we only train the proposal.

**Evaluation** In the right panel of Figure 1, we compare the bound gaps of NAS-X and NASMC averaged across 20 different samples from the generative model. To obtain the bound gap for NAS-X, we run SMC 16 times with 128 particles and with the learned proposal and twists. We then record the average log marginal likelihood. For NASMC, we run SMC with the current learned proposal (without any twists).

## 10 rSLDS

**Model details** The generative model is as follows. At each time $t$, there is a discrete latent state $z_t \in \{1, \ldots, 4\}$ as well as a two-dimensional continuous latent state $x_t \in \mathbb{R}^2$. The discrete state

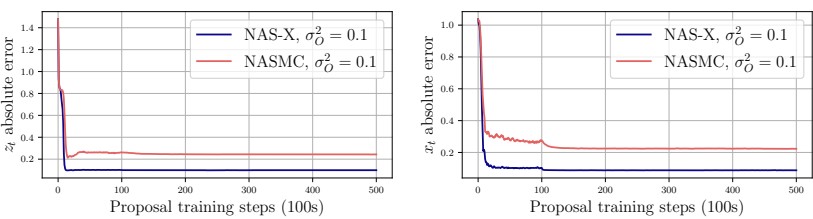

Figure 6: **Inference in NASCAR experiments.**

transition probabilities are given by

$$p(z_{t+1} = i \mid z_t = j, x_t) \propto \exp\left(r_i + R_i^T x_{t-1}\right) \tag{39}$$

Here $R_i$ and $r_i$ are weights for the discrete state $z_i$.

These discrete latent states dictates two-dimensional latent state $x_t \in \mathbb{R}^2$ which evolves according to linear Gaussian dynamics.

$$x_{t+1} = A_{z_{t+1}} x_t + b_{z_{t+1}} + v_t, \qquad v_t \sim^{\text{iid}} \mathcal{N}(0, Q_{z_{t+1}}) \tag{40}$$

Here $A_k, Q_k \in \mathbb{R}^{2x2}$ and $b_k \in \mathbb{R}^2$. Importantly, from Equations 40 and 39 we see that the dynamics of the continuous latent states and discrete latents are coupled. The discrete latent states index into specific linear dynamics and the discrete transition probabilities depend on the continuous latent state.

The observations $y_t \in \mathbb{R}^{10}$ are linear projections of the continuous latent state $x_t$ with some additive Gaussian noise.

$$y_t = Cx_t + d + w_t, \qquad v_t \sim^{\text{iid}} \mathcal{N}(0, S) \tag{41}$$

Here $C, S \in \mathbb{R}^{10x10}$ and $d \in \mathbb{R}^{10}$.

**Proposal Parameterization**    We use a mean-field proposal distribution factorized over the discrete and continuous latent variables (i.e. $q(\mathbf{z}_{1:T}, \mathbf{x}_{1:T}) = q(\mathbf{z}_{1:T})q(\mathbf{x}_{1:T})$). For the continuous states, $q(\mathbf{x}_{1:T})$ is a Gaussian factorized over time with parameters $\mu_{1:T}$ and $\sigma^2_{1:T}$. For the discrete states, $q(\mathbf{z}_{1:T})$ is a Categorical distribution over $K$ categories factorized over time with parameters $p^{1:K}_{1:T}$. In total, we learn $2T + TK$ proposal parameters.

**Twist Parameterization**    We parameterize the twists using a recurrent neural network (RNN) that is trained using density ratio estimation. To produce the twist values at each timestep, we first run a RNN backwards over the observations $\mathbf{y}_{1:T}$ to produce a sequence of encodings $\mathbf{e}_{1:T-1}$. We then concatenate the encodings of $\mathbf{x}_t$ and $\mathbf{z}_t$ into a single vector and pass that vector into an MLP which outputs the twist values at each timestep. The RNN has one layer with 128 hidden units. The MLP has 131 hidden units and ReLU activations.

**Model Parameter Evaluation**    We closely follow the parameter initialization strategy employed by Linderman et al. [41]. First, we use PCA to obtain a set of continuous latent states and initialize the matrices $C$ and $d$. We then fit an autoregressive HMM to the estimated continuous latent states in order to initialize the dynamics matrices $\{A_k, b_k\}$. Importantly, we do not initialize the proposal with the continuous latent states described above.

**Training Details**    We use a batch size of 32 for the density ratio estimation step. We alternate between 100 steps of twist training and 100 steps of proposal training for a total of 50,000 training steps in total. We used Adam and considered a grid search over the model, proposal, and twist learning rates. In particular, we considered learning rates of $1e-4, 1e-3, 1e-2$ for the model, proposal, and twist.

**Bootstrap Bound Evaluation**    To obtain the log marginal likelihood bounds and standard deviations in Table 5.2, we ran a bootstrapped particle filter (BPF) with the learned model parameters for all three methods (NAS-X, NASMC, Laplace EM) using 1024 particles. We repeat this across 30 random seeds. Initialization of the latent states was important for a fair comparison. To initialize the latent

states, for NAS-X and NASMC, we simply sampled from the learned proposal at time $t = 0$. To initialize the latent state for Laplace EM, we sampled from a Gaussian distribution with the learned dynamics variance at $t = 0$.

**Inference Comparison**    In the top panel of Figure 6, we compare NAS-X and NASMC on inference in the SLDS model. We report (average) posterior parameter recovery for the continuous and discrete latent states across 5 random samples from the generative model. NAS-X systematically recovers better estimates of both the discrete and continuous latent states.

# 11    Inference in Squid Giant Axon Model

## 11.1    HH Model Definition

For the inference experiments (Section 5.3.1) we used a probabilistic version of the squid giant axon model [9, 43]. Our experimental setup was constructed to broadly match [19], and used a single-compartment model with dynamics defined by

$$C_m \frac{dv}{dt} = I_{\text{ext}} - \bar{g}_{\text{Na}} m^3 h(v - E_{\text{Na}}) - \bar{g}_{\text{K}} n^4 (v - E_{\text{K}}) - g_{\text{leak}}(v - E_{\text{leak}}) \tag{42}$$

$$\frac{dm}{dt} = \alpha_m(v)(1 - m) - \beta_m(v)m \tag{43}$$

$$\frac{dh}{dt} = \alpha_h(v)(1 - h) - \beta_h(v)h \tag{44}$$

$$\frac{dn}{dt} = \alpha_n(v)(1 - n) - \beta_n(v)n \tag{45}$$

where $C_m$ is the membrane capacitance; $v$ is the potential difference across the membrane; $I_{\text{ext}}$ is the external current; $\bar{g}_{\text{Na}}$, $\bar{g}_{\text{K}}$, and $\bar{g}_{\text{leak}}$ are the maximum conductances for sodium, potassium, and leak channels; $E_{\text{Na}}$, $E_{\text{K}}$, and $E_{\text{leak}}$ are the reversal potentials for the sodium, potassium, and leak channels; $m$ and $h$ are subunit states for the sodium channels and $n$ is the subunit state for the potassium channels. The functions $\alpha$ and $\beta$ that define the dynamics for $n$, $m$, and $h$ are defined as

$$\alpha_m(v) = \frac{-4 - v/10}{\exp(-4 - v/10) - 1}, \quad \beta_m(v) = 4 \cdot \exp((-65 - v)/18) \tag{46}$$

$$\alpha_h(v) = 0.07 \cdot \exp((-65 - v)/20), \quad \beta_h(v) = \frac{1}{\exp(-3.5 - v/10) + 1} \tag{47}$$

$$\alpha_n(v) = \frac{-5.5 - v/10}{\exp(-5.5 - v/10) - 1}, \quad \beta_n(v) = 0.125 \cdot \exp((-65 - v)/80) \tag{48}$$

This system of ordinary differential equations defines a nonlinear dynamical system with a four-dimensional state space: the instantaneous membrane potential $v$ and the ion gate subunit states $n$, $m$, and $h$.

As in [19], we use a probabilistic version of the original HH model that adds zero-mean Gaussian noise to both the membrane voltage $v$ and the "unconstrained" subunit states. The observations are produced by adding Gaussian noise with variance $\sigma_y^2$ to the membrane potential $v$.

Specifically, let $\mathbf{x}_t$ be the state vector of the system at time $t$ containing $(v_t, m_t, h_t, n_t)$, and let $\varphi_{dt}(\mathbf{x})$ be a function that integrates the system of ODEs defined above for a step of length $dt$. Then the probabilistic HH model can be written as

$$p(\mathbf{x}_{1:T}, \mathbf{y}_{1:T}) = p(\mathbf{x}_1) \prod_{t=2}^{T} p(\mathbf{x}_t \mid \varphi_{dt}(\mathbf{x}_{t-1})) \prod_{t=1}^{T} \mathcal{N}(\mathbf{y}_t; \mathbf{x}_{t,1}, \sigma_y^2) \tag{49}$$

where the 4-D state distributions $p(\mathbf{x}_1)$ and $p(\mathbf{x}_t \mid \varphi_{dt}(\mathbf{x}_{t-1}))$ are defined as

$$p(\mathbf{x}_t \mid \varphi_{dt}(\mathbf{x}_{t-1})) = \mathcal{N}(\mathbf{x}_{t,1}; \varphi_{dt}(\mathbf{x}_{t-1})_1, \sigma_{x,1}^2) \prod_{i=2}^{4} \text{LogitNormal}(\mathbf{x}_{t,i}; \varphi_{dt}(\mathbf{x}_{t-1})_i, \sigma_{x,i}^2). \tag{50}$$

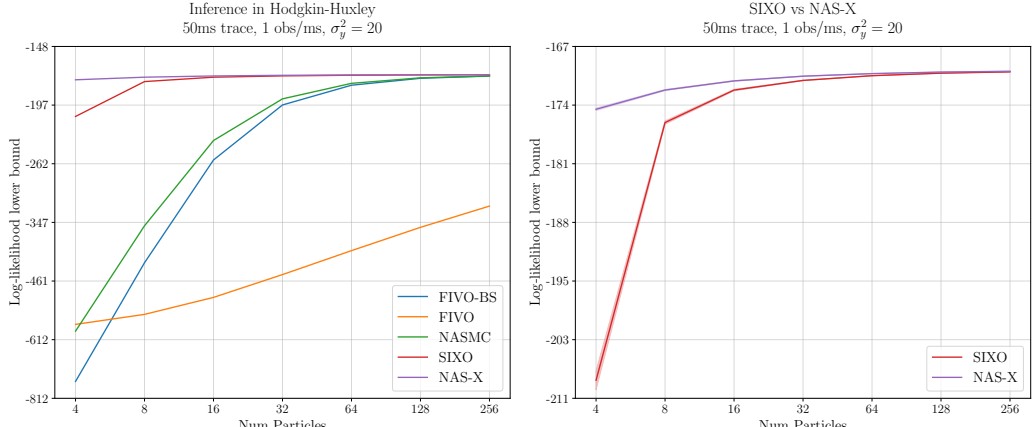

Figure 7: **HH inference performance across different numbers of particles.**
**(left)** Log-likelihood lower bounds for proposals trained with 4 particles and evaluated across a range of particle numbers. NAS-X's inference performance decays only minimally as the number of particles is decreased, while all other methods experience significant performance degradation.
**(right)** A comparison of SIXO and NAS-X containing the same values as the left panel, but zoomed in. NAS-X is roughly twice as particle efficient as SIXO, and outperforms SIXO by roughly 34 nats at 4 particles.

In words, we add Gaussian noise to the voltage ($\mathbf{x}_{t,1}$) and logit-normal noise to the gate states $n, m$, and $h$. The logit-normal is defined as the distribution of a random variable whose logit has a Gaussian distribution, or equivalently it is a Gaussian transformed by the sigmoid function and renormalized. We chose the logit-normal because its values are bounded between 0 and 1, which is necessary for the gate states.

**Problem Setting** For the inference experiments we sampled 10,000 noisy voltage traces from a fixed model and used each method to train proposals (and possibly twists) to compute the marginal likelihood assigned to the data under the true model.

As in [19], we sampled trajectories of length 50 milliseconds, with a single noisy voltage observation every millisecond. The stability of our ODE integrator allowed us to integrate at $dt = 0.1$ms, meaning that there were 10 latent states per observation.

**Proposal and Twist Details** Each proposal was parameterized using the combination of a bidirectional recurrent neural network (RNN) that conditioned on all observed noisy voltages as well as a dense network that conditioned on the RNN hidden state and the previous latent state $\mathbf{x}_{t-1}$ [45, 49]. The twists for SIXO and NAS-X were parameterized using an RNN run in reverse over the observations combined with a dense network that conditioned on the reverse RNN hidden state and the latent being 'twisted', $\mathbf{x}_t$. Both the proposal and twists were learned in an amortized manner, i.e. they were shared across all trajectories. All RNNs had a single hidden layer of size 64, as did the dense networks. All models were fit with ADAM [50] with proposal learning rate of $10^{-4}$ and twist learning rate of $10^{-3}$.

A crucial aspect of fitting the proposals was defining them in terms of a 'residual' from the prior, a technique known as Res$_q$ [7]. In our setting, we defined the true proposal density as proportional to the product of a unit-variance Gaussian centered at $\varphi(\mathbf{x}_t)$ and a Gaussian with parameters output from the RNN proposal.

### 11.2 Experimental Results

In Figure 7 we plot the performance of proposals and twists trained with 4 particles and evaluated across a range of particle numbers. All methods except FIVO perform roughly the same when evaluated with 256 particles, but with lower numbers of evaluation particles the smoothing methods emerge as more particle-efficient than the filtering methods. To achieve NAS-X's inference perfor-

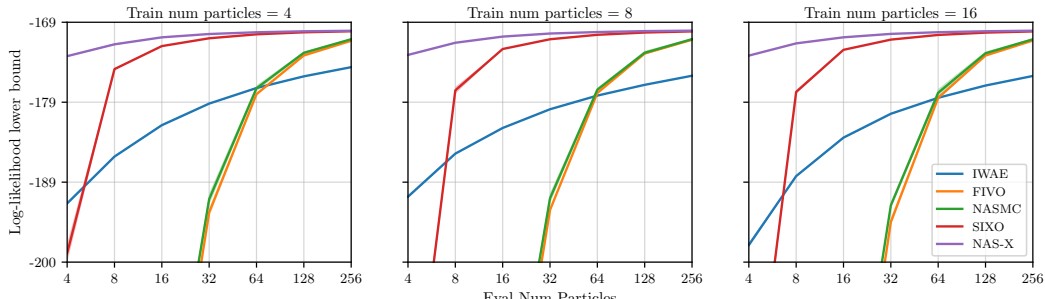

Figure 8: **Training HH proposals with increasing numbers of particles.** HH proposal performance plots similar to Figure 7, but trained with varying numbers of particles. Increasing the number of particles at training time has a negligible effect on NAS-X performance in this setting, but caused many VI-based methods to perform worse. This could be due to signal-to-noise issues in proposal gradients, as discussed in Rainforth et al. [51].

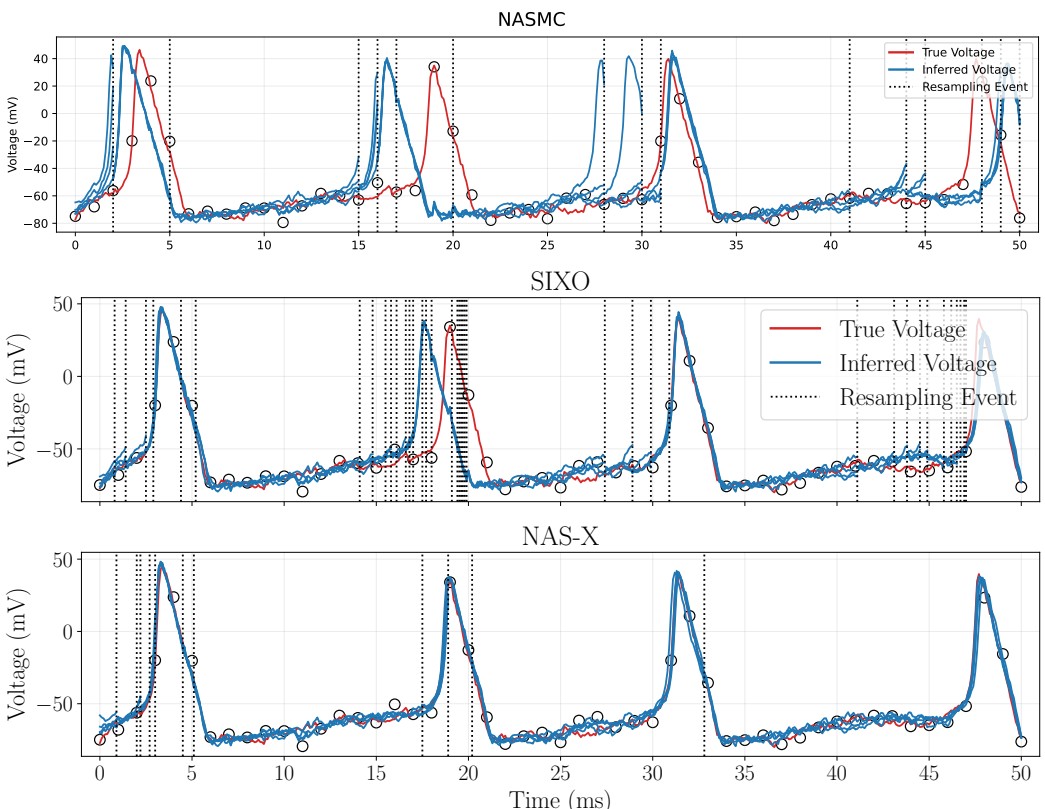

Figure 9: **Inferred voltage traces for NASMC, SIXO, and NAS-X.**
**(top)** NASMC exhibits poor performance, incorrectly inferring the timing of most spikes. **(middle)** SIXO's inferred voltage traces are more accurate than NASMC's with only a single mistimed spike, but SIXO generates a high number of resampling events leading to particle degeneracy. **(bottom)** NAS-X perfectly infers the latent voltage with no mistimed spikes, and resamples very infrequently.

mance with 4 particles, NASMC would need 256 particles, a 64-times increase. NAS-X is also more particle-efficient than SIXO, achieving on average a 2x particle efficiency improvement. We show the effect of changing the number of training particles in Figure 8.

The FIVO method with a parametric proposal drastically underperformed all smoothing methods as well as NASMC, indicating that the combination of filtering SMC and the exclusive KL divergence leads to problems optimizing the proposal parameters. To compensate, we also evaluated the performance of "FIVO-BS", a filtering method that uses a bootstrap proposal. This method is identical to a bootstrap particle filter, i.e. it proposes from the model and has no trainable parameters. FIVO-BS far outperforms standard FIVO, and is only marginally worse than NASMC in this setting.

In Figure 9 we investigate these results qualitatively by examining the inferred voltage traces of each method. We see that NASMC struggles to produce accurate spike timings and generates many spurious spikes, likely because it is unable to incorporate future information into its proposal or resampling method. SIXO performs better than NASMC, accurately inferring the timing of most spikes but resampling at a high rate. High numbers of resampling events can lead to particle degeneracy and poor inferences. NAS-X is able to correctly infer the voltage across the whole trace with no spurious or mistimed spikes. Furthermore NAS-X rarely resamples, indicating it has learned a high-quality proposal that does not generate low-quality particles that must be resampled away. These qualitative results seem to support the quantitative results in Figure 7 — SIXO's high resampling rate and NASMC's filtering approach lead to lower bound values.

Table 1: **Train Bound comparison**

| Metric | NAS-X | SIXO |
|---|---|---|
| $\mathcal{L}_{\text{BPF}}^{256}$ | $-660.7003$ | $-636.2579$ |
| $\mathcal{L}_{\text{train}}^{4}$ | $-664.3528$ | $-668.6865$ |
| $\mathcal{L}_{\text{train}}^{8}$ | $-662.8712$ | $-653.6352$ |
| $\mathcal{L}_{\text{train}}^{16}$ | $-662.0753$ | $-644.8764$ |
| $\mathcal{L}_{\text{train}}^{32}$ | $-661.5387$ | $-639.5388$ |
| $\mathcal{L}_{\text{train}}^{64}$ | $-660.8040$ | $-636.5131$ |
| $\mathcal{L}_{\text{train}}^{128}$ | $-660.5102$ | $-633.7875$ |
| $\mathcal{L}_{\text{train}}^{256}$ | $-660.3423$ | $-632.1377$ |

## 12 Model Learning in Mouse Pyramidal Neuron Model

### 12.1 Model Definition

For the model learning experiments in Section 5.3.2 we used a generalization of the Hodgkin-Huxley model developed for modeling mouse visual cortex neurons by the Allen Institute for Brain Science [48, 52]. Specifically we used the perisomatic model with ID 482657528 developed to model cell 480169178. The model is detailed in the whitepaper [52] and the accompanying code, but we reproduce the details here to ensure our work is self-contained.

Similar to the squid giant axon model, the mouse visual cortex model is composed of ion channels that affect the current flowing in and out of the cell. Let $\mathcal{I}$ be the set of ions $\{\text{Na}^+, \text{Ca}^{2+}, \text{K}^+\}$. Each ion has associated with it

1. A set of channels that transport that ion, denoted $C_i$ for $i \in \mathcal{I}$.
2. A reversal potential, $E_i$.
3. An instantaneous current density, $I_i$, which is computed by summing the current density flowing through each channel that transports that ion.

Correspondingly, let $\mathcal{C}$ be the set of all ion channels so that $\mathcal{C} = \bigcup_{i \in \mathcal{I}} C_i$. Each $c \in \mathcal{C}$ has associated with it

1. A maximum conductance density, $\overline{g}_c$.
2. A set of subunit states, referred to collectively as the vector $\lambda_c$. Let $\lambda_c \in [0, 1]^{d_c}$, i.e. $\lambda_c$ is a $d_c$-dimensional vector of values in the interval $[0, 1]$.

3. A function $g_c$ that combines the gate values to produce a number in $[0, 1]$ that weights the maximum conductance density, $\overline{g}_c \cdot g_c(\lambda_c)$.

4. Functions $A_c(\cdot)$ and $b_c(\cdot)$ which compute the matrix and vector used in the ODE describing $\lambda_c$ dynamics. $A_c$ and $b_c$ are functions of both the current membrane voltage $v$ and calcium concentration inside the cell $[Ca^{2+}]_i$. If the number of subunits (i.e. the dimensionality of $\lambda_c$) is $d_c$, then the output of $A_c(v, [Ca^{2+}]_i)$ is a $d_c \times d_c$ diagonal matrix and the output of $b_c(v, [Ca^{2+}]_i)$ is a $d_c$-dimensional vector.

With this notation we can write the system of ODEs

$$C_m \frac{dv}{dt} = \frac{I_{\text{ext}}}{SA} - g_{\text{leak}}(v - E_{\text{leak}}) - \sum_{i \in \text{ions}} I_i \tag{51}$$

$$I_i = \sum_{c \in C_i} \overline{g}_c g_c(\lambda_c)(v - E_i) \tag{52}$$

$$\frac{d\lambda_c}{dt} = A_c(v, [Ca^{2+}]_i)\lambda_c + b_c(v, [Ca^{2+}]_i) \quad \forall c \in C \tag{53}$$

$$\frac{d[Ca^{2+}]_i}{dt} = -kI_{Ca^{2+}} - \frac{[Ca^{2+}]_i - [Ca^{2+}]_{\text{min}}}{\tau}. \tag{54}$$

Most symbols are as described earlier, $SA$ is the membrane surface area of the neuron, $[Ca^{2+}]_i$ is the calcium concentration inside the cell, $[Ca^{2+}]_{\text{min}}$ is the minimum interior calcium concentration with a value of 1 nanomolar, $\tau$ is the rate of removal of calcium with a value of 80 milliseconds, and $k$ and is a constant with value

$$k = 10000 \cdot \frac{\gamma}{2 \cdot F \cdot \text{depth}} \tag{55}$$

where 10000 is a dimensional constant, $\gamma$ is the percent of unbuffered free calcium, $F$ is Faraday's constant, and $\text{depth}$ is the depth of the calcium buffer with a value of 0.1 microns.

Because the concentration of calcium changes over time, this model calculates the reversal potential for calcium $E_{Ca^{2+}}$ using the Nernst equation

$$E_{Ca^{2+}} = \frac{G \cdot T}{2 \cdot F} \log\left(\frac{[Ca^{2+}]_o}{[Ca^{2+}]_i}\right) \tag{56}$$

where $G$ is the gas constant, $T$ is the temperature in Kelvin (308.15°), $F$ is Faraday's constant, and $[Ca^{2+}]_o$ is the extracellular calcium ion concentration which was set to 2 millimolar.

**Probabilistic Model**    The probabilistic version of the deterministic ODEs was constructed similarly to the probabilistic squid giant axon model — Gaussian noise was added to the voltage and unconstrained gate states. One difference is that the system state now includes $[Ca^{2+}]_i$ which is constrained to be greater than 0. To noise $[Ca^{2+}]_i$ we added Gaussian noise in the log space, analogous to the logit-space noise for the gate states.

**Model Size**    The 38 learnable parameters of the model include:

1. Conductances $\overline{g}$ for all ion channels (10 parameters).

2. Reversal potentials of sodium, potassium, and the non-specific cation: $E_{K^+}$, $E_{Na^+}$, and $E_{NSC^+}$.

3. The membrane surface area and specific capacitance.

4. Leak channel reversal potential and max conductance density.

5. The calcium decay rate and free calcium percent.

6. Gaussian noise variances for the voltage $v$ and interior calcium concentration $[Ca^{2+}]_i$.

7. Gaussian noise variances for all subunit states (16 parameters).

8. Observation noise variance.

The 18-dimensional state includes:

1. Voltage $v$
2. Interior calcium concentration $[\text{Ca}^{2+}]_i$
3. All subunit states (16 dimensions)

## 12.2 Channel Definitions

In this section we provide a list of all ion channels used in the model. In the following equations we often use the function $\mathrm{exprel}$ which is defined as

$$
\mathrm{exprel}(x) = \begin{cases} 1 & \text{if} \quad x = 0 \\ \dfrac{\exp(x) - 1}{x} & \text{otherwise} \end{cases} \tag{57}
$$

A numerically stable implementation of this function was critical to training our models.

Additionally, many of the channel equations below contain a 'temperature correction' $q_t$ that adjusts for the fact that the original experiments and Allen Institute experiments were not done at the same temperature. In those equations, $T$ is the temperature in Celsius which was $35°$.

### 12.2.1 Transient Na$^+$

From Colbert and Pan [53].

$$
\lambda_c = (m, h), \quad g_c(\lambda_c) = m^3 h
$$

$$
\frac{1}{q_t}\frac{dm}{dt} = \alpha_m(v)(1 - m) - \beta_m(v)m
$$

$$
\frac{1}{q_t}\frac{dh}{dt} = \alpha_h(v)(1 - h) - \beta_h(v)h
$$

$$
q_t = 2.3^{\left(\frac{T-23}{10}\right)}
$$

$$
\alpha_m(v) = \frac{0.182 \cdot 6}{\mathrm{exprel}(-(v + 40)/6)}, \quad \beta_m(v) = \frac{0.124 \cdot 6}{\mathrm{exprel}((v + 40)/6)}
$$

$$
\alpha_h(v) = \frac{0.015 \cdot 6}{\mathrm{exprel}((v + 66)/6)}, \quad \beta_h(v) = \frac{0.015 \cdot 6}{\mathrm{exprel}(-(v + 66)/6)}
$$

### 12.2.2 Persistent Na$^+$

From Magistretti and Alonso [54].

$$
\lambda_c = h, \quad g_c(\lambda_c) = m_\infty h
$$

$$
m_\infty = \frac{1}{1 + \exp(-(v + 52.6)/4.6)}
$$

$$
\frac{1}{q_t}\frac{dh}{dt} = \alpha_h(v)(1 - h) - \beta_h(v)h
$$

$$
q_t = 2.3^{\left(\frac{T-21}{10}\right)}
$$

$$
\alpha_h(v) = \frac{2.88 \times 10^{-6} \cdot 4.63}{\mathrm{exprel}((v + 17.013)/4.63)}, \quad \beta_h(v) = \frac{6.94 \times 10^{-6} \cdot 2.63}{\mathrm{exprel}(-(v + 64.4)/2.63)}
$$

### 12.2.3 Hyperpolarization-activated cation conductance

From Kole et al. [55]. This channel uses a 'nonspecific cation current' meaning it can transport any cation. In practice, this is modeled by giving it its own special ion $\text{NSC}^+$ with resting potential

$E_{\text{NSC}^+}$.

$$\lambda_c = m, \quad g_c(\lambda_c) = m$$
$$E_{\text{NSC}^+} = -45.0$$
$$\frac{dm}{dt} = \alpha_m(v)(1-m) - \beta_m(v)m$$
$$\alpha_m(v) = \frac{0.001 \cdot 6.43 \cdot 11.9}{\text{exprel}((v+154.9)/11.9)}, \quad \beta_m(v) = 0.001 \cdot 193 \cdot \exp(v/33.1)$$

### 12.2.4 High-voltage-activated Ca²⁺ conductance

From Reuveni et al. [56]

$$\lambda_c = (m,h), \quad g_c(\lambda_c) = m^2 h$$
$$\frac{dm}{dt} = \alpha_m(v)(1-m) - \beta_m(v)m$$
$$\frac{dh}{dt} = \alpha_h(v)(1-h) - \beta_h(v)h$$
$$\alpha_m(v) = \frac{0.055 \cdot 3.8}{\text{exprel}(-(v+27)/3.8)}, \quad \beta_m(v) = 0.94 \cdot \exp(-(v+75)/17)$$
$$\alpha_h(v) = 0.000457 \cdot \exp(-(v+13)/50), \quad \beta_h(v) = \frac{0.0065}{\exp(-(v+15)/28)+1}$$

### 12.2.5 Low-voltage-activated Ca²⁺ conductance

From Avery and Johnston [57], Randall and Tsien [58].

$$\lambda_c = (m,h), \quad g_c(\lambda_c) = m^2 h$$
$$\frac{1}{q_t}\frac{dm}{dt} = \frac{m_\infty - m}{m_\tau}$$
$$\frac{1}{q_t}\frac{dh}{dt} = \frac{h_\infty - h}{h_\tau}$$
$$q_t = 2.3^{(T-21)/10}$$
$$m_\infty = \frac{1}{1+\exp(-(v+40)/6)}, \quad m_\tau = 5 + \frac{20}{1+\exp((v+35)/5)}$$
$$h_\infty = \frac{1}{1+\exp((v+90)/6.4)}, \quad h_\tau = 20 + \frac{50}{1+\exp((v+50)/7)}$$

### 12.2.6 M-type (Kv7) K⁺ conductance

From Adams et al. [59].

$$\lambda_c = m, \quad g_c(\lambda_c) = m$$
$$\frac{1}{q_t}\frac{dm}{dt} = \alpha_m(v)(1-m) - \beta_m(v)m$$
$$q_t = 2.3^{\left(\frac{T-21}{10}\right)}$$
$$\alpha_m(v) = 0.0033\exp(0.1(v+35)), \quad \beta_m(v) = 0.0033 \cdot \exp(-0.1(v+35))$$

### 12.2.7 Kv3-like K⁺ conductance

$$\lambda_c = m, \quad g_c(\lambda_c) = m$$
$$\frac{dm}{dt} = \frac{m_\infty - m}{m_\tau}$$
$$m_\infty = \frac{1}{1+\exp(-(v-18.7)/9.7)}, \quad m_\tau = \frac{4}{1+\exp(-(v+46.56)/44.14)}$$

### 12.2.8 Fast inactivating (transient, Kv4-like) K$^+$ conductance

From Korngreen and Sakmann [60].

$$\lambda_c = (m, h), \quad g_c(\lambda_c) = m^4 h$$

$$\frac{1}{q_t}\frac{dm}{dt} = \frac{m_\infty - m}{m_\tau}$$

$$\frac{1}{q_t}\frac{dh}{dt} = \frac{h_\infty - h}{h_\tau}$$

$$q_t = 2.3^{(T-21)/10}$$

$$m_\infty = \frac{1}{1 + \exp(-(v+47)/29)}, \quad m_\tau = 0.34 + \frac{0.92}{\exp(((v+71)/59)^2)}$$

$$h_\infty = \frac{1}{1 + \exp((v+66)/10)}, \quad h_\tau = 8 + \frac{49}{\exp(((v+73)/23)^2)}$$

$$\bar{g} = 1 \times 10^{-5}$$

### 12.2.9 Slow inactivating (persistent) K$^+$ conductance

From Korngreen and Sakmann [60].

$$\lambda_c = (m, h), \quad g_c(\lambda_c) = m^2 h$$

$$\frac{1}{q_t}\frac{dm}{dt} = \frac{m_\infty - m}{m_\tau}$$

$$\frac{1}{q_t}\frac{dh}{dt} = \frac{h_\infty - h}{h_\tau}$$

$$q_t = 2.3^{(T-21)/10}$$

$$m_\infty = \frac{1}{1 + \exp(-(v+14.3)/14.6)}$$

$$m_\tau = \begin{cases} 1.25 + 175.03 \cdot e^{0.026v}, & \text{if } v < -50 \\ 1.25 + 13 \cdot e^{-0.026v}, & \text{if } v \geq -50 \end{cases}$$

$$h_\infty = \frac{1}{1 + \exp((v+54)/11)}$$

$$h_\tau = \frac{24v + 2690}{\exp(((v+75)/48)^2)}$$

$$\bar{g} = 1 \times 10^{-5}$$

### 12.2.10 SK-type calcium-activated K$^+$ conductance

From Köhler et al. [61]. Note this is the only calcium-gated ion channel in the model.

$$\lambda_c = z, \quad g_c(\lambda_c) = z$$

$$\frac{dz}{dt} = \frac{z_\infty - z}{z_\tau}$$

$$z_\infty = \frac{1}{1 + (0.00043/[\text{Ca}^{2+}]_i)^{4.8}}, \quad z_\tau = 1$$

## 12.3 Training Details

**Dataset**   The dataset used to fit the model was a subset of the stimulus/response pairs available from the Allen Institute. First, all stimuli and responses were downloaded for cell 480169178. Then, sections of length 200 milliseconds were extracted from a subset of the stimuli types. The stimuli types and sections were chosen so that the neuron was at rest and unstimulated at the beginning of the trace. We list the exclusion criteria below.

1. Any "Hold" stimuli: Excluded because these traces were collected under voltage clamp conditions which we did not model.

2. Test: Excluded because the stimulus is 0 mV for the entire trace.

3. Ramp/Ramp to Rheobase: Excluded because the cell is only at rest at the very beginning of the trace.

4. Short Square: 250 ms to 450 ms.

5. Short Square — Triple: 1250 ms to 1450 ms.

6. Noise 1 and Noise 2: 1250 ms to 1450 ms, 9250 ms to 9450 ms, 17250 ms to 17450 ms.

7. Long Square: 250 ms to 450 ms.

8. Square — 0.5ms Subthreshold: The entire trace.

9. Square — 2s Suprathreshold: 250 ms to 450 ms.

10. All others: Excluded.

For cell 480169178, the criteria above selected 95 stimulus/response pairs of 200 milliseconds each. Each trace pair was then downsampled to 1 ms (from the original 0.005 ms per step) and corrupted with mean-zero Gaussian noise of variance 20 mV$^2$ to simulate voltage imaging conditions. Finally, the 95 traces were randomly split into 72 training traces and 23 test traces.

**Proposal and Twist**   The proposal and twist hyperparameters were broadly similar to the squid axon experiments, with the proposal being parameterized by a bidirectional RNN with a single hidden layer of size 64 and an MLP with a single hidden layer of size 64. The RNN was conditioned on the observed response and stimulus voltages at each timestep, and the MLP accepted the RNN hidden state, the previous latent state, and a transformer positional encoding of the number of steps since the last voltage response observation. The twist was similarly parameterized using an RNN run in reverse across the stimulus and response, combined with an MLP that accepted the RNN hidden state, the latent state being evaluated, and a transformer positional encoding of the number of steps elapsed since the last voltage response observation. The positional encodings were used to inform the twist and proposal of the number of steps elapsed since the last observation because the model was integrated with a stepsize of 0.1ms while observations were received once every millisecond.

**Hyperparameter Sweeps**   To evaluate the methods we swept across the parameters

1. Initial observation variance: $e^2, e^3, e^5$

2. Initial voltage dynamics variance: $e, e^2, e^3$

3. Bias added to scales produced by the proposal: $e^2, e^5$

We also evaluated the models across three different data noise variances (20, 10, and 5) but the results were similar for all values, so we reported only the results for variance 20. This amounted to $3 \cdot 3 \cdot 3 \cdot 2$ different hyperparameter settings, and 5 seeds were run for each setting yielding a total of 270 runs.

When computing final performance, a hyperparameter setting was only evaluated if it had at least 3 runs that achieved 250,000 steps without NaN-ing out. For each hyperparameter setting selected for evaluation, all successful seeds were evaluated using early stopping on the train 4-particle log likelihood lower bound.

## 13   Strang Splitting for Hodgkin-Huxley Models

Because the Hodgkin-Huxley model is a *stiff* ODE, integrating it can be a challenge, especially at large step sizes. The traditional solution is to use an implicit integration scheme with varying step size, allowing the algorithm to take large steps when the voltage is not spiking. However, because our model adds noise to the ODE state at each timestep adaptive step-size methods are not viable as the different stepsizes would change the noise distribution.

Instead, we sought an explicit, fixed step-size method that could be stably integrated at relatively large stepsizes. Inspired by Chen et al. [47], we developed a splitting approach that exploits the conditional linearity of the system. The system of ODEs describing the model can be split into two subsystems

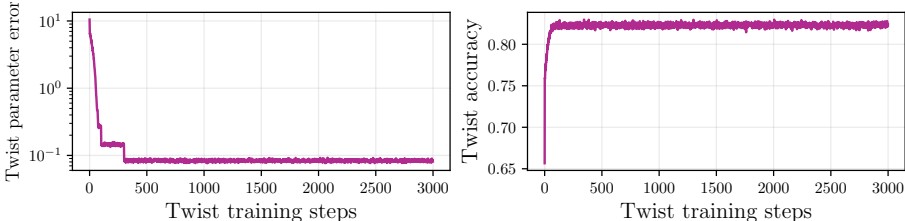

Figure 10: **Twist learning in LG-SSM. (left)** Twist parameter error relative to optimal twist parameters for LG-SSM task; **(right)** Classification accuracy of learned twist. With an appropriate twist parameterization, twist learning via density ratio estimation is robust.

of linear first-order ODEs when conditioned on the state of the other subsystem. Specifically, the dynamics of the channel subunit states $\{\lambda_c \mid c \in \mathcal{C}\}$ is a system of linear first-order ODEs when conditioned on the voltage $v$ and interior calcium concentration $[\text{Ca}^{2+}]_i$. Similarly, the dynamics for $v$ and $[\text{Ca}^{2+}]_i$ is a system of linear first-order ODEs when conditioned on the subunit states.

Because the conditional dynamics of each subsystem are linear first-order ODEs, an exact solution to each subsystem is possible under the assumption that the states being conditioned on are constant for the duration of the step. Our integration approach uses these exact updates in an alternating fashion, first performing an exact update to the voltage and interior calcium concentration while holding the subunit states constant, and then performing an exact update to the subunit states while holding the voltage and interior calcium concentration constant. For details on Strang and other splitting methods applied to Hodgkin-Huxley type ODEs, see [47].

## 14 Robustness of Twist Learning

NAS-X uses SIXO's twist learning framework to approximate the smoothing distributions. The twist learning approaches involves density ratio estimation. In brief, the density ratio estimation procedure involves training a classifier to distinguish between samples from $p_{\boldsymbol{\theta}}(\mathbf{x}_t \mid \mathbf{y}_{t+1:T})$ and $p_{\boldsymbol{\theta}}(\mathbf{x}_t)$. These samples can be obtained from the generative model. For details see Section 2.2. In principle, incorporating twists complicates the overall learning problem and traditional methods for twist learning can indeed be challenging. However, in practice, twist learning using the SIXO framework is robust and easy. In Figure 10, we present twist parameter recovery and classification accuracy for the Gaussian SSM experiments (Section 5.1); in this setting, the optimal twists have a known parametric form. The optimal twist parameters are recovered quickly, the classification accuracy is high, and training is stable. This suggests that, with an appropriate twist parameterization, twist learning via density ratio estimation is tractable and straightforward.

## 15 Computational Complexity and Wall-clock Time

Theoretically, all multi-particle methods considered (NAS-X, SIXO, FIVO, NASMC, RWS, IWAE) have $O(KT)$ time complexity, where $K$ is the number of particles and $T$ is the number of time steps. Once concern is that the resampling operation in SMC could require super-linear time in the number of particles, but drawing $K$ samples from a $K$-category discrete distribution can be accomplished in $O(K)$ time using the alias method [62, 63]. Additionally, for NAS-X, evaluating the twists is amortized across timesteps as in Lawson et al. [19], giving time linear in $T$.

NAS-X and SIXO have similar wall-clock speeds but are slower than FIVO and NASMC, primarily because of twist training, see Table 2. Even if FIVO and NASMC were run with more particles to equalize wall-clock times, they would still far underperform NAS-X in log marginal likelihood lower bounds, see Figure 7.

Specifically, SIXO and NAS-X take 3.5x longer per step than NASMC and FIVO and 2.5x longer per step than RWS and IWAE. However, Figure 7 shows that FIVO, NASMC, IWAE, and RWS cannot match NAS-X's performance even with 64 times more computation (4 vs. 256 particles). SIXO only matches NAS-X's performance with 4x as many particles (4 vs. 16 particles). Therefore, NAS-X uses computational resources much more effectively than other methods.

| Method | ms / global step | ms / proposal step | ms / twist step |
|--------|------------------|--------------------|-----------------|
| IWAE   | $70.3 \pm 15.9$  | $70.3 \pm 15.9$    | N/A             |
| RWS    | $71.6 \pm 8.2$   | $71.6 \pm 8.2$     | N/A             |
| NASMC  | $53.9 \pm 6.6$   | $53.9 \pm 6.6$     | N/A             |
| FIVO   | $51.4 \pm 13.3$  | $51.4 \pm 13.3$    | N/A             |
| NAS-X  | $163.2 \pm 39.8$ | $73.5 \pm 18.5$    | $89.7 \pm 21.3$ |
| SIXO   | $175.3 \pm 42.4$ | $85.6 \pm 21.1$    | $89.7 \pm 21.3$ |

Table 2: **Wall-clock speeds of various methods during HH inference.**

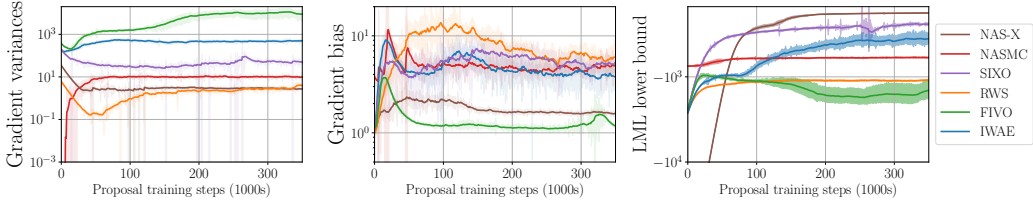

Figure 11: Hodgkin-Huxley gradient variances **(left)** gradient bias **(middle)**, and log-marginal likelihood lower bounds **(right)** over training.

## 16 Empirical analysis of bias and variance of gradients

In Figure 11, we analyze the gradient variance and bias for the Hodgkin-Huxley experiments, supplementing our theoretical analyses in Section 3.1. Figure 11 (left) shows NAS-X attains lower variance gradient estimates than IWAE, FIVO, and SIXO with comparable variance to RWS. We also studied the bias (middle) by approximating the true gradient by running a bootstrap particle filter with 256 particles using the best proposal from the inference experiments. NAS-X's gradients are lower bias than all methods but FIVO, but FIVO's gradients are also the highest variance. We hypothesize that FIVO's gradients appear less biased because its parameters are pushed towards degenerate values where gradient estimation is "easier". We illustrate this in the right panel, where we plot log-marginal likelihood bounds.

