# OpenReview forum: "NAS-X: Neural Adaptive Smoothing via Twisting"
_NeurIPS.cc/2023/Conference — NeurIPS 2023 poster_

### Official Review · Reviewer_mkCV · 2023-07-03

**Soundness:** 3 good
**Presentation:** 3 good
**Contribution:** 2 fair
**Rating:** 6
**Confidence:** 4

**Summary:**

The paper proposes NAS-X, a novel approach to estimate the posterior expectations in Reweighted Wake Sleep (RWS) architectures. Instead of a traditional estimation using self-normalized importance sampling (SNIS), and similar to Neural Adaptive Sequential Monte-Carlo (NASMC), the proposed method employs a Sequential Monte-Carlo (SMC) approach to estimate the necessary expectations. In contrast to NASMC, NAS-X uses smoothing distributions instead of filtering distributions as targets, which improves particle efficiency and reduces the variance of the estimates. This requires the estimation of twist sequences, which the paper does using the density-ratio approach SIXO.

**Strengths:**

- Efficient particle-based estimation of posteriors in sequence models is an important topic and has a long history in the machine learning community.

- The paper is well-written and relatively easy to follow. I was particularly impressed by the paper’s natural motivation: starting with a concise recap of RWS, the paper clearly explains the need for smoothing SMC and its estimation via twists.

- The main technical contribution (Eq.(8) and Eq.(9)) is SIXO-based smoothing SMC for posterior inference in RWS architectures. From a technical point of view this contribution is relatively simple, because all necessary pieces (RWS, NASMC, and SIXO) were already available, but the insight that the RWS updates are compatible with SIXO estimation is noteworthy and must be appreciated.

- The experiments validate the proposed approach in settings with known ground truth and discrete latent variables, as well as challenging inference in Hodgkin-Huxley models. While the experiments with Gaussian-linear models and Switching Linear Dynamical Systems are important sanity checks and confirm the advantages of smoothing SMC over filtering SMC, I particularly enjoyed the real-world experiments on voltage dynamics in neural membranes. They not only demonstrate that NAS-X is more particle-efficient than its competitors but also that the proposed model could be useful beyond synthetic environments.

**Weaknesses:**

- My main concern with this paper is its potential impact: SMC-based inference in RWS architectures is already a niche topic and there is nothing fundamentally wrong with filtering SMC. Smoothing SMC introduces the additional complication of twist estimation, but even that has already been addressed with SIXO. My worry is that the remaining contribution likely has a relatively small audience.

- Another point of concern are NAS-X’s multi-level approximations: SMC is already approximate in nature, but now, in addition to the quality and efficiency of the particles, accurate estimation of the twist sequence becomes a separate challenge. There is a trade-off between the additional information contained in future observations and a decrease in robustness due to a more complex learning problem, and I would have liked to see experiments that directly evaluate the quality of the learned twists.

- The paper claims that smoothing SMC leads to lower-variance estimates compared to filtering SMC (l.32f, l.92f), but the experiments do not investigate this claim directly. Some indirect evidence is provided in Figure 1 and Figure 3, but I would have appreciated additional insights.

- The description of the experimental setup could be better structured. I did find most of the information I was looking for, but it required constant jumping between different sections of the paper and between the paper and the supplemental material. The information contained in the supplemental material is also not referenced well enough in the main paper and I would encourage the authors to include more pointers to the supplemental material.

**Questions:**

- The paper generally does a good job giving credit to prior work, but the relationship between SIXO and NAS-X is not clear enough. If SIXO already performs smoothing SMC, what is the main difference between the two? Is NAS-X more than SIXO applied to RWS? What architecture is used for SIXO *without* NAS-X?

- The effect of the number of training particles on the reported performance metrics remains a bit of a mystery. I would appreciate a reproduction of Figure 3(a) with 32 and 128 training particles.

- The meaning of “inclusive KL”, in contrast to “exclusive KL”, is not clear enough. I found the definitions in the literature, but they should be mentioned in the paper.

Typos: l.48.5 (“,”), l.52 (“goals”).

**Limitations:**

- The paper does not discuss the limitations of the proposed model. Completely absent in this work is a runtime analysis. Does NAS-X's higher particle efficiency translate to faster (absolute wall time) inference compared to NASMC/SIXO?

- The paper does not discuss the potential negative societal impact of their work.

---

> ### Author Rebuttal · Authors · 2023-08-10
>
> Thanks for your comments and feedback. Just to be clear on terminology --- In your review you refer to “RWS architectures”, but we weren’t sure if you meant model architecture or a general framework for developing new methods. We take the later interpretation and refer to it as the “RWS approach” in our response.
>
> **My main concern with this paper is its potential impact**
>
> Thank you for this feedback, proper positioning of the work is extremely important!
>
> First, we underscore the broad applicability of our work. NAS-X is a method for model learning and inference in Markovian state-space models (SSMs). Model learning and inference in SSMs is an important topic to NeurIPS and the broader machine learning community, with many applications (e.g., neuroscience, healthcare, deep generative models) [1,2,3,4,5,6,7,8].
>
> In addition, we believe NAS-X is of broader methodological interest than SIXO, as it outperforms SIXO on most tasks and can fit discrete latent variable models which SIXO cannot. NAS-X is not limited in its applicability by using RWS; instead, it leverages RWS- and SIXO-inspired methods to tackle broader problems than SIXO or RWS can on their own.
>
> In many problems, we agree there is nothing wrong with filtering SMC. However, in some problems, smoothing is crucial (e.g., the Hodgkin-Huxley model). We showed that NAS-X performs better on the HH inference task with 4 particles than FIVO or NASMC does with 256 particles, a 64x improvement over filtering SMC. Our theoretical results also support this --- FIVO and NASMC’s gradient estimates are not consistent, whereas NAS-X’s gradient estimates are consistent. While there is nothing wrong with filtering SMC, in many settings, there can be strong gains from using smoothing techniques.
>
> [1] Krishnan, Rahul G. et al. “Structured Inference Networks for Nonlinear State Space Models.” AAAI (2016).
>
> [2] Fox, Emily B. et al. “Nonparametric Bayesian Learning of Switching Linear Dynamical Systems.” NeurIPS (2008).
>
> [3] Costacurta, Julia C., et al. "Distinguishing Discrete and Continuous Behavioral Variability Using Warped Autoregressive HMMs." NeurIPS (2022).
>
> [4] Alaa, Ahmed M. et al. “Attentive State-Space Modeling of Disease Progression.” NeurIPS (2019).
>
> [5] Ghahramani, Zoubin et al. “Variational Learning for Switching State-Space Models.” Neural Computation 12 (2000).
>
> [6] Miller, Andrew C. et al. “Learning Insulin-Glucose Dynamics in the Wild.” MLHC (2020)
>
> [7] Chung, Junyoung et al. “A Recurrent Latent Variable Model for Sequential Data.” NeuIPS (2015).
>
> [8] Wu, Luhuan et al. “Practical and Asymptotically Exact Conditional Sampling in Diffusion Models.” (2023)
>
> **The relationship between SIXO and NAS-X is not clear enough**
>
> We view the RWS approach as ascending estimates of the gradients of the log marginal likelihood (LML) of a latent variable model. These gradients are expectations wrt the posterior over latents and can be approximated using biased but consistent estimates from self-normalized importance sampling (SNIS).
>
> Changing the SNIS method gives new model-fitting techniques. Thus, NASMC uses filtering SMC (also an SNIS method) to estimate gradients, and NAS-X uses smoothing SMC. NAS-X approximates smoothing SMC using twists learned via density ratio estimation, as in SIXO.
>
> This contrasts the VI approach, which ascends unbiased estimates of a lower bound on the LML. FIVO uses filtering SMC’s estimate of the marginal likelihood to define a lower bound on the LML, and SIXO uses smoothing SMC similarly.
>
> The crucial difference is that RWS-based methods follow biased but consistent estimates of the gradients of the true LML, while VI-based methods follow unbiased estimates of lower bounds of the LML. These different approaches result in gradient estimators with different strengths and weaknesses.
>
> Both NAS-X and SIXO use (approximate) smoothing SMC as part of their algorithms. The difference is that NAS-X uses the samples from smoothing SMC to estimate the gradients in a way that allows it to outperform SIXO and be more broadly applicable, as supported by our empirical results.
>
> **Another point of concern are NAS-X’s multi-level approximations**
>
> NAS-X's approximations provide important theoretical and practical advantages. For example, NASMC incurs bias in the gradient estimates because it approximates the posterior distribution with filtering distributions. We view introducing the twists not as a further approximation but as an attempt to fix NASMC's approximation. Our experiments and additional comparisons show NAS-X performs favorably against all baselines, providing evidence that this change is helpful in practice.
>
> In principle, incorporating twists complicates the learning problem. In practice, twist learning is robust and easy. In the PDF, we present twist parameter recovery and classification accuracy for the Gaussian SSM experiments; in this setting, the optimal twists have a known parametric form. The optimal twist parameters are recovered quickly, the classification accuracy is high, and training is stable. This suggests that, with an appropriate twist parameterization, twist learning via density ratio estimation is tractable. Twist learning was stable and successful in all other experiments as well, we'll include full results in the revision.
>
> **The paper claims that smoothing SMC leads to lower-variance estimates**
>
> In new results, we show that NAS-X has lower variance and lower bias gradient estimates than filtering SMC-based methods (see general response).
>
> This claim is well-established in the literature (e.g. FIVO). Furthermore, as discussed above, the advantage of smoothing SMC in the context of NASMC is also to reduce bias of gradient estimates.
>
> **Effect of number of training particles...**
>
> We didn't have the resources to reproduce Figure 3a for 32/128 particles; see PDF for 8 and 16 particles.
>
> **We are constrained by space, please see general response for the rest of your concerns.**

---

> > ### Comment · Reviewer_mkCV · 2023-08-14
> >
> > I want to thank the authors for their insightful response, in particular the clarifications about the differences between NAS-X and SIXO, the inclusion of additional baselines, and new experiments analyzing NAS-X's gradients. I still believe that the paper proposes a relatively low-level fix that introduces non-trivial complexity and, while I appreciate the authors' comments about twist stability and robustness, I think there are many open questions regarding their parametrization and recovery in real-world settings.

---

> > > ### Author Response · Authors · 2023-08-14
> > >
> > > Thank you for your response! Do you feel that we addressed your concerns enough to merit an increase in our score? You said your main concern was the impact of the method compared to the complexity required. We tried to clarify in our response that the usefulness and impact of NAS-X exceed that of SIXO for an algorithm of exactly the same complexity (both technical and computational). Stated differently, NAS-X is a drop-in replacement for SIXO that only requires changing a few lines of code but performs significantly better and can handle discrete latents. Regardless of your choice, thank you for your feedback and discussion!

---

> > > > ### Comment · Reviewer_mkCV · 2023-08-20
> > > >
> > > > Thank you for these clarifications. Based on this discussion I've decided to increase the rating to 6.

---

### Official Review · Reviewer_DvQm · 2023-07-04

**Soundness:** 3 good
**Presentation:** 3 good
**Contribution:** 3 good
**Rating:** 6
**Confidence:** 3

**Summary:**

The paper tackles the problem of learning sequential latent variable models using a new method called NAS-X. It combines two previous research: 1. The smoothing with twisting that’s applied in variational inference (SIXO in particular). It approximates smoothing distribution as targets for proposal learning, instead of using filtering distribution, to avoid sample degeneracy; 2. The Reweighted wake-sleep method, which can handle discrete latent variables. They used experiments to illustrate that NAS-X can learn proposals that match the true posterior marginals, and can be applied to discrete variables and handle complex tasks.



**Strengths:**

The paper is well written. The authors have done extensive experiments to show the effectiveness of their work. The proposed method is somewhat novel, as it combines two existing ideas and outperforms them.

**Weaknesses:**

The paper lacks discussions of future work and limitations of the proposed method.

**Questions:**

1. Section 2.1: I find the structure there a bit unclear. you first described a two-step coordinate ascent method in the first paragraph, but then it is unclear whether your formular afterwards are talking about the first or the second step, it might make sense to be clearer there.

2. Section 2.2 SMC description: the “repeats three steps” paragraph doesn’t make it clear that it is repeatedly sampling the next time point (if I understand correctly). Maybe it is better to just be clear that each step is for a new t.

3. Section 5.2: it is unclear to me that the qualitative comparison in figure 2 brings that much insight, but maybe it at least shows how the data is generated?

4. Section 5.3.1: for figure 3(a) you showed that is It more particle efficient, I wonder if it would be helpful to also comment on the computation efficiency there?

5. Section 5.3.1 last paragraph: The discussion about RWS vs VI seems intriguing but I wish it can be discussed a bit more clearly and thoroughly and provide more context. Also if we believe this to be an important enough point, maybe it is worth highlighting this point in the intro or conclusion?

6. Figure 4: Why would we expect NAS-X to outperform in certain metrics but underperform in other metrics?

7. Section 5.3.2: I wonder why NASMC gets dropped from the results in this section?



**Limitations:**

I wonder if the authors could add more discussions on the limitations/future work of their proposed method?

---

> ### Author Rebuttal · Authors · 2023-08-10
>
> We thank the reviewer for their comments and constructive feedback. Below we respond to the questions/concerns raised.
>
> **The paper lacks discussions of future work and limitations of the proposed method.**
>
> One limitation is the NAS-X does not have a unified objective (i.e., the proposal and model are updated using different objectives). On the other hand, both FIVO and SIXO use a unified objective, namely SMC’s lower bound on the log-marginal likelihood.  While this had no practical impact on our experiments, the lack of a unified objective could be relevant in other applications.
>
> Another limitation is that SMC’s gradient estimates are consistent but biased; empirically, this doesn’t seem to prevent NAS-X from recovering proposal and model parameters in settings where we know those parameters. However, we think a thorough theoretical analysis of how the bias of the gradients depends on the quality of the twist approximation and number of particles is an exciting topic for future work.
>
> **Section 5.3.1: for figure 3(a) you showed that is It more particle efficient, I wonder if it would be helpful to also comment on the computation efficiency there?**
>
> See general response
>
> **Section 5.3.1 last paragraph: The discussion about RWS vs VI seems intriguing but I wish it can be discussed a bit more clearly and thoroughly and provide more context**
>
> We can expand upon this in more detail in the revision but elaborate here briefly. We find that VI methods such as SIXO tend to learn more entropic proposals. We saw this in the Hodgkin-Huxley experiments and the Gaussian state-space model experiments.
> We think this property is related to alternative interpretations of the VI methods. These VI methods were originally motivated as using Monte Carlo algorithms to derive tighter lower bounds on the log-marginal likelihood, hopefully resulting in a better learning objective.
>
> However, an alternative interpretation is that these VI methods maximize a standard ELBO objective but with an “expanded” variational family. The expanded family is defined by *both* the chosen proposal family (q) and the Monte Carlo algorithm. As a concrete example, you can interpret the IWAE bound as maximizing the ELBO with a proposal defined by this procedure:
>
> 1. Draw K samples x_1, … , x_k from q(x | y).
> 2. Weight the K samples using the density ratio p(x_i, y) / q(x_i | y)
> 3. Draw a sample from the set of weighted samples.
>
> The IWAE bound can therefore be thought of as maximizing the ELBO but with a variational family that allows the underlying proposal (q) to propose K samples and choose the “best” one. This perspective is also explored in the paper "Energy Inspired Models" (Lawson et al. 2019).
>
> Intuitively, if the proposal (q) in IWAE has K chances to draw a high-quality sample, it is incentivized to be more entropic, since only the best sample is chosen. A similar argument holds for bounds based on more complex Monte Carlo algorithms like FIVO and SIXO. Samples from the underlying proposal (q) drive Monte Carlo algorithms that select only the best particles. This allows the proposal to generate many dubious samples provided it generates at least one good sample.
>
> Crucially, this interpretation does not hold for RWS-based algorithms (RWS, NASMC, NAS-X). RWS algorithms use self-normalized importance sampling variants like filtering and smoothing SMC to directly estimate the gradients of the log marginal likelihood. The Monte Carlo algorithms are only used to improve the gradient estimates, and cannot be interpreted as augmenting the proposal distribution. In practice, this results in proposals that are not as entropic as VI-trained proposals because they are not “aware” that their bad particles will be resampled away.
>
> We hope this clarifies things and are happy to discuss more.
>
> **Figure 4: Why would we expect NAS-X to outperform in certain metrics but underperform in other metrics?**
>
> We are not entirely sure why NAS-X would outperform on some metrics and underperform on others, but suspect it could be related to the less entropic proposals it learns. We provided both standard model learning metrics and biophysical metrics because different users may value different metrics. For example, our biophysical metrics were motivated by the ones used in [1].
>
> This is similar to other fields like image modeling, where it has become standard practice to evaluate model samples on perceptual metrics like Frechet inception distance (FID) as well as likelihoods. This is because many models that achieve the highest likelihoods did not generate good samples, and many models with the best samples did not provide tractable likelihood evaluation (e.g., GANS).
>
> [1] Lueckmann, J.M et al. Flexible statistical inference for mechanistic models of neural dynamics. NeurIPS (2017).
>
> **Section 5.3.2: I wonder why NASMC gets dropped from the results in this section?**
>
> We found HH model learning under the IWAE, NASMC, RWS, and FIVO objectives to be highly unstable. Few runs survived to converge, instead NaN-ing out early. For example, 46 percent of all IWAE runs failed to converge, compared to 19 percent of NAS-X runs and 25 percent of SIXO runs. This prevented us from reliably evaluating the methods, and we concluded that they were ill-suited for the model learning task. We were able to successfully evaluate all of these methods for proposal learning (see new figures), however, and found them to be far inferior to SIXO and NAS-X.

---

> > ### Comment · Reviewer_DvQm · 2023-08-18
> >
> > Thanks the authors for their detailed response! They have addressed my questions.

---

> > > ### Author Response · Authors · 2023-08-18
> > >
> > > Glad we could clarify things! Do you feel our additional results and explanation merit changing your rating or confidence score?

---

### Official Review · Reviewer_nTEg · 2023-07-06

**Soundness:** 3 good
**Presentation:** 3 good
**Contribution:** 2 fair
**Rating:** 5
**Confidence:** 4

**Summary:**

An algorithm based on reweighted wake-sleep (RWS) is applied to sequential latent variable models $p_\theta(x_{1:T}, y_{1:T})$. Model parameters are fit by maximizing the evidence $p_\theta(y_{1:T})$ in $\theta$, and posterior inference is performed by minimizing the forward KL divergence $KL(p_\theta(x_{1:T} \mid y_{1:T}) \mid \mid q_\phi(x_{1:T} \mid \mid y_{1:T}))$ in variational parameters $\phi$. Gradients for each of these tasks are estimated by smoothing SMC, which can be viewed as a lower-variance alternative to self-normalized importance sampling (SNIS) in the sequential setting. This submission extends the filtering SMC approach of previous work. However, the smoothing distributions are not available and must be approximated by learning twists, which is accomplished by training a discriminator. This results in an estimate of an appropriate density ratio that allows for twisting to be used.


**Strengths:**

* The exposition is clear the and potential advantages with respect to forward KL minimization and smoothing are intuitive.
* The method is general and and appears to be applicable in any learning/inference setting with sequential latent variable models. The only significant design choice for the user appears to be the design of the variational distribution, and the additional computational overhead from the training of the classifier for estimating the twists is not prohibitive.
* The method is novel to the best of my knowledge, and combining a forward KL minimization with SIXO seems like a good idea. The forward KL objective makes it easy to handle discrete latent variables, as the second experiment shows, whereas other methods (see below) could not do so as easily.
* The authors show the applicability of the method to three experiments of increasing complexity, with an emphasis on dynamical systems.


**Weaknesses:**

The biggest weaknesses of the paper are:
* Lack of comparison to existing methods. In addition to NASMC, FIVO (Maddison et al., 2017) and VSMC (Naesseth et al., 2018), and AESMC (Le et al.,  2018) are closely related methods that instead target the reverse KL divergence. While these are cited in related work, they should be compared to when possible (perhaps not in discrete latent variable models, though) as they are natural competitors for these sequential latent variable models.
* Lack of discussion of, and comparison to, other SMC variants. The main intuition provided in the paper for using smoothing SMC is that both SNIS and filtering SMC can result in either high-variance estimates and/or particle degeneracy. Metrics (e.g. effective sample size) and figures for validating this intuition would be helpful, as this intuition is the central motivation for the use of smoothing SMC.

Ablation studies incorporating these competitors/alternatives would make more clear the importance of the contribution of this work. At present, it is difficult to tell whether both the smoothing approach or the forward KL formulation contribute to the success of the proposed method, or whether in some cases just one of these does.


**Questions:**

* Was vanilla RWS compared to? Naive implementations wouldn’t use sequential structure, and would rely only on SNIS, so I’d expect all of NASMC, SIXO, and NAS-X to beat RWS easily. Nevertheless, it would be a nice baseline.
* Can there be more discussion on the classifier being used to approximate the density ratio? Some precise results from the GAN literature or the literature on likelihood free inference by ratio estimation (Thomas et al., 2016) stating formally that the optimal classification rule yields the likelihood ratio in some form would make this part of the exposition more clear; at least some citations should be added.
* To what extent are the advantages of the proposed method from 1) use of the forward KL divergence and 2) use of smoothing SMC?
 Comparison with FIVO, VSMC, AESMC along with naive RWS or ELBO could really help make this clear.
* Is the $\chi$ in the title on purpose? Or should there be an X as in the rest of the body? It should be consistent.
* On Open Review, the “smothing” in the title should corrected if possible.


**Limitations:**

The authors have made clear the class of probabilistic models that this method is designed for.

---

> ### Author Rebuttal · Authors · 2023-08-10
>
> We thank the reviewer for their comments and feedback. In addition to the small changes you suggested, we also respond to your main questions and concerns:
>
> **Lack of comparison to existing methods.**
>
> In new experiments (see PDF), we compare NAS-X to several more baselines, including FIVO, SIXO, RWS, IWAE, and ELBO. In short, NAS-X outperforms or performs comparably to these baselines in all three experiments. These additional results establish that 1) we outperform methods that employ reverse KL + smoothing while being able to handle discrete latent variables (e.g., SIXO), 2) we outperform both forward KL and reverse KL-based methods that use filtering SMC (e.g., NASMC, FIVO) and 3) we outperform several baselines that ignore sequential structure (RWS, IWAE, ELBO). Additionally, in new empirical results, we show that NAS-X has lower variance and lower bias gradient estimates than these methods (see general response).
>
> Altogether, our new empirical results highlight the benefits of using smoothing SMC in conjunction with a forward-KL divergence objective for sequential latent variable models.
>
> We did not include IWAE, FIVO, or SIXO in the discrete latent experiment (rSLDS) because (as you imply) they rely on continuous reparameterization gradients. Instead, we compared to Laplace EM, a very strong baseline that analytically marginalizes out the latents.
>
> **Lack of discussion of, and comparison to, other SMC variants… motivation for the use of smoothing SMC**
>
> First, to address your concerns we included an analysis of variance and bias of gradient estimates for the Hodgkin-Huxley experiments. This includes comparisons to methods based on SNIS (e.g., IWAE, RWS) and filtering SMC (e.g., FIVO, NASMC). In short (see general response), NAS-X’s use of smoothing SMC leads to lower variance gradient estimates and lower bias.
>
> Second, while this is certainly problem-dependent, the claim that filtering SMC’s estimates can be high variance is relatively well-established in the literature [1,2,3,4,5]. Performing reliable inference and model learning in problems where smoothing information is highly important (e.g. neural voltage data) was a primary motivation for our work.
>
> In addition to the practical motivations, we wish to highlight theoretical motivations for using smoothing SMC in the context of RWS: 1) NAS-X’s gradient estimates are consistent when the twists are optimal, 2) NAS-X produces unbiased gradient estimates even for finite particles, provided that the twists and proposal are optimal. Please see the general response for a detailed discussion. Importantly, NASMC does not have these guarantees because it uses filtering SMC’s intermediate particle approximations to estimate expectations wrt the true posterior, to avoid particle degeneracy. Since these intermediate particles approximate the filtering distributions and NOT the true posterior, this introduces additional bias into gradient estimates; this bias persists even in the infinite particle limit. We illustrated this in Experiment 1, where NASMC did not recover the true posterior. In contrast, NAS-X’s gradients are unbiased, in theory, when the twists and proposal are optimal. We illustrated this advantage in the Gaussian state-space model experiments, and will provide a discussion/formal proof of this result in the revision to motivate our method better.
>
> [1] Lawson, Dieterich et al. “Twisted Variational Sequential Monte Carlo.” (2018).
>
> [2] Lawson, Dieterich et al. “SIXO: Smoothing Inference with Twisted Objectives.” (2022).
>
> [3] Naesseth, Christian Andersson et al. “Elements of Sequential Monte Carlo.” Found. Trends Mach. Learn. 12 (2019): 307-392.
>
> [4] Heng, Jeremey et al. “Controlled Sequential Monte Carlo” (2019).
>
> [5] Guarniero, Pieralberto et al. "The iterated auxiliary particle filter." Journal of the American Statistical Association 112.520 (2017): 1636-1647.
>
> **Ablation studies incorporating these competitors [...] would make more clear the importance of the contribution of this work**
>
> Thanks for this suggestion. We wholeheartedly agree. In new ablation studies (see general response) we observe, broadly speaking, that the RWS-based methods (RWS, NASMC, NAS-X) provide lower-variance gradients and learn less-entropic proposals than their variational inference counterparts (ELBO, FIVO, SIXO). Lower-variance gradients are an important advantage as long as they are also low-bias, and our experiments show that learning the twists allows NAS-X to provide the lowest-bias gradient estimates of all methods considered. Thus, both the KL direction and the twist learning contribute to NAS-X’s performance.
> In addition to the performance benefits of NAS-X over its competitors, the forward-KL methods are also more versatile because they can accommodate discrete latent variables.
>
> **Was vanilla RWS compared to?**
>
> We have included comparisons to RWS and other non-sequential baselines (e.g., ELBO, IWAE). We find that in all experiments, NAS-X significantly outperforms RWS.
>
>  **Can there be more discussion on the classifier being used to approximate the density ratio?**
>
> We will discuss this in more detail in the revised version. For experiment 1, we chose a parametric form that matches the analytic density ratio for the Gaussian state-space model. For experiments 2 and 3, we use RNNs that take the observations in as inputs and produce encodings for each time step. We feed these encodings and the latent states into an MLP classifier for each time step. We’ll include the citations below and expand our discussion of SIXO’s density ratio approach in the revision; please suggest any additional citations.
>
> [1] Sugiyama, Masashi et al. Density ratio estimation in machine learning. Cambridge University Press, 2012.
>
> [2] Thomas, Owen et al. “Likelihood-Free Inference by Ratio Estimation.” Bayesian Analysis (2016).
>
> [3] Uehara, Masatoshi et al. “Generative Adversarial Nets from a Density Ratio Estimation Perspective.” arXiv (2016).

---

> > ### Comment · Reviewer_nTEg · 2023-08-16
> > **Rebuttal reply**
> >
> > Thanks to the authors for the detailed response.
> >
> > 1) The inclusion of FIVO as a competitor has suitably addressed my concern about limited evaluation. In the Gaussian SSM, it's clear that NAS-X outperforms FIVO. Although I think this is a simple example (and thus that it may be important to contextualize this by noting that learning the twists is (perhaps) more straightforward because of the simplicity), it's a fine toy example and shows a setting where NAS-X is clearly stronger than FIVO. The comparison of bias and variance of gradients is also a nice addition, whether it ends up in the main body or supplement. I'm updating my score from 4 to 5 to reflect these additions.
> >
> > 2) The "new theoretical results" alluded to by the authors in the global and individuals responses may be a nice touch, but ultimately may have limited utility: any result that relies on the condition "when the twists are optimal" seems to assume away the extremely complicated task of learning the twists (as acknowledged by the authors on line 102). While these may provide some nice motivation or intuition, in more complicated models beyond Gaussian SSM this may not be a fair assumption.

---

> > > ### Author Response · Authors · 2023-08-16
> > >
> > > Thanks for your response and engaging with our rebuttal! We really appreciate you taking the time.
> > >
> > > In regards to your first point, we wanted to emphasize that we also compared to FIVO in the HH rebuttal experiments and found it to be far inferior to NAS-X. For example, NAS-X with 4 particles achieved a log marginal likelihood lower bound of -173 nats while FIVO with 4 particles achieved -579 nats and FIVO with 128 particles achieved -173 nats. So FIVO only matched NAS-X using 32 times as many particles. Plots of these values are in the rebuttal PDF (Figure 4 left). Unfortunately, FIVO Nan-ed out too frequently to be reliably evaluated for HH model learning. We believe this is a significant non-toy example where NAS-X clearly outperforms FIVO.
> > >
> > > As a side note, we were also excited to see that NAS-X outperformed FIVO for the LGSSM as that is a setting where filtering methods are generally seen as sufficient.
> > >
> > > For your second point, in our practical experience, twist learning via DRE is quite easy and robust. We provide experimental evidence of this in the rebuttal PDF for the LGSSM, but all experiments were the same in this regard. We are happy to provide plots of twist performance for the other experiments if that would help.
> > >
> > > We apologize for any confusion in the unclear wording of line 102. When we said “Learning the twists can be extremely challenging” we were referring to previous methods that use a Bellman-type loss to learn the twists (iAPF [1], cSMC [2], and TVSMC [3]). These methods struggle or completely fail in complex, high dimensional settings. Indeed, those struggles motivated our choice of DRE for twist learning and our experience is that it works very well. Even if the true twists are never recovered exactly we still observed orders of magnitude performance gains over not using the twists, which implies they are useful even when not 100% correct.
> > >
> > > Thanks again for your valuable feedback!
> > >
> > > [1] Guarniero, Pieralberto, et al. "The iterated auxiliary particle filter."
> > >
> > > [2] Heng, Jeremy, et al. "Controlled sequential Monte Carlo."
> > >
> > > [3] Lawson, Dieterich, et al. "Twisted variational sequential Monte Carlo."

---

### Official Review · Reviewer_Dwhv · 2023-07-07

**Soundness:** 3 good
**Presentation:** 3 good
**Contribution:** 3 good
**Rating:** 6
**Confidence:** 2

**Summary:**

The authors propose to use SIXO particle approximation to calculate the expectations in reweighted wake-sleep algorithm for state space models.

**Strengths:**

The paper is written in a clear way and all required background is well-explained (but description of  SIXO and twists is too short).

The idea of replacing the variational expectations with best possible practical approximation of the posterior is interesting.

The empirical findings are well-presented.


**Weaknesses:**

I would lengthen the description of SIXO and twists.

More empirical evidence and discussion why NAS-X actually warrants better performance (as it introduces biases) would strengthen the paper.

Some theoretical analysis would strengthen the paper.

**Questions:**

What do you think are main limitations of NAS-X?

When the algorithm is expected to perform poorly (or be outperformed by Laplace EM)?

---

> ### Author Rebuttal · Authors · 2023-08-10
>
> We thank the reviewer for their positive comments and constructive feedback. Below we respond to the questions/concerns raised.
>
> **I would lengthen the description of SIXO and twists.**
>
> Thank you for the suggestion. We have added a  detailed discussion of the technical details behind SIXO and twists in the revised version of the paper.
>
> **More empirical evidence and discussion why NAS-X actually warrants better performance.**
>
> In new experiments (see PDF), we compare NAS-X to several more baselines, including FIVO, SIXO, RWS, IWAE, and ELBO. In short, NAS-X either outperforms or performs comparably to these baselines in all experiments. These additional results establish that 1) we outperform methods that employ reverse KL + smoothing while being able to handle discrete latents (e.g., SIXO) 2) we outperform both forward KL and reverse KL-based methods that use filtering SMC (e.g., NASMC, FIVO) and 3) we outperform several baselines that ignore sequential structure (RWS, IWAE, ELBO). Additionally, in new empirical results, we show that NAS-X has lower variance and lower bias gradient estimates than these methods (see general response).
>
> Altogether, our new empirical results highlight the benefits of using smoothing SMC in conjunction with a forward-KL divergence objective for sequential latent variable models.
>
> **Some theoretical analysis would strengthen the paper.**
>
> We will include two theoretical guarantees in the revised paper: 1) NAS-X’s gradient estimates are consistent when the twists are optimal 2) NAS-X produces unbiased gradient estimates even for a finite number of particles, provided that the twists and proposal are optimal. Please see the general response for a detailed discussion of this.
>
> Importantly, NASMC does not have these guarantees because it uses filtering SMC’s intermediate particle approximations to estimate expectations w.r.t. the posterior. Since these intermediate particles approximate the filtering distributions and NOT the smoothing distributions, this introduces additional bias into gradient estimates. Importantly, this bias persists even in the infinite particle limit. From this perspective, you can view the twists not as introducing bias but as compensating for the bias inherent in NASMC’s approach.
>
> **What do you think are main limitations of NAS-X?**
>
> One limitation is the NAS-X does not have a unified objective (i.e., the proposal and model are updated using different objectives). In certain settings, this could lead to a divergence of model and proposal parameters.
>
> FIVO does have a unified objective for the model and proposal, namely SMC’s lower bound on the log-marginal likelihood.  SIXO shares this objective for the model and proposal but learns the twist using the separate density ratio estimation objective, as in NAS-X.
>
> While this had no practical impact on our experiments, the lack of a unified objective could be relevant in other applications.
>
> Another limitation is that SMC’s gradient estimates are consistent but biased for finite numbers of particles in practice (when twists and proposals are imperfect). We think a thorough theoretical analysis of how the bias of the gradients depends on the quality of the twist approximation and the number of particles is an exciting topic for future work.
>
> A final limitation is that NAS-X only works in the offline setting where all observations are available. It cannot be used for streaming data.
>
> **When is the algorithm expected to perform poorly (or be outperformed by Laplace EM)?**
>
> Because NAS-X is a sampling-based method, we would not expect it to perform as well as methods that analytically marginalize out discrete latent variables, such as Laplace EM [1]. However, analytic marginalization is only tractable when the number of latent variables is small. Our rSLDS experiments were designed to address this directly and show NAS-X matching or slightly outperforming Laplace EM even in a model with 4 discrete states. This shows that our sampling-based method performs reasonably even in settings where it is at a disadvantage.
>
> [1] Zoltowski, David, Jonathan Pillow, and Scott Linderman. "A general recurrent state space framework for modeling neural dynamics during decision-making." Proceedings of the 37th International Conference on Machine Learning, vol. 119, PMLR, 2020

---

> > ### Comment · Reviewer_Dwhv · 2023-08-19
> >
> > I thank the authors for the response which addresses my questions. I maintain my score. I encourage the authors to explicitly discuss the limitations in the manuscript.

---

### Author Rebuttal · Authors · 2023-08-10

We thank the reviewers for their detailed feedback. We respond to reviewers individually and provide a general response below. We have strengthened our submission with several new experiments (see figures in PDF) and theoretical analyses (see below). If the reviewers feel that the new experimental results, analyses, and clarifications resolve their concerns, we kindly ask them to consider updating their overall scores. We are happy to provide further clarifications.

### New Experimental Results
#### Additional Baselines
In new experiments, we show NAS-X outperforms alternative methods suggested by the reviewers (ELBO, IWAE, FIVO, SIXO, and RWS). NAS-X’s improvements over these methods highlight 1) the benefits of using smoothing SMC versus filtering SMC and self-normalized importance sampling and 2) the benefits of forward KL over reverse KL in the context of sequential latent variable models. Results include:
* Figure 1. Gaussian State Space Model: NAS-X achieves a tighter lower bound and lower proposal parameter error than SIXO, FIVO, ELBO, IWAE, NASMC, and RWS.
* Figure 3. r-SLDS: NAS-X attains a higher bootstrap particle filter bound than NASMC and RWS. SIXO and FIVO rely on continuous reparameterization gradients and so were not compared to.
* Figure 4. Hodgkin-Huxley Inference experiments: We include new comparisons between IWAE, FIVO, NASMC, SIXO, and NAS-X for proposals trained with 4, 8, and 16 particles. These show that NAS-X substantially outperforms all other methods even as the number of training particles changes. NAS-X achieves the same performance as SIXO with 4x fewer particles, and achieves the same performance as FIVO and NASMC with 64x fewer particles. RWS results were also obtained but were too poor to fit on the plots.

#### Robustness of twist learning
* Figure 2 illustrates that twist learning is robust, plotting convergence to true twist parameters and twist classification accuracy in the Gaussian SSM.

### Computational Complexity/Wall Clock Time
Several reviewers asked about computational complexity. We provide a summary of our results here, and will include a full discussion in the final version of the paper.

Theoretically, all methods have O(KT) time complexity, where K is the number of particles and T is the number of time steps.
Practically, NAS-X and SIXO have similar wall-clock times but are slower than FIVO and NASMC, primarily because of twist training.
Even if FIVO and NASMC were run with more particles to equalize wall-clock times, they would still far underperform NAS-X in log marginal likelihood lower bounds.

In the accompanying PDF, we include a table (Figure 6) detailing the wall-clock speed of each method in milliseconds per step during HH inference training runs.

SIXO and NAS-X take ~3.5x longer per step than NASMC and FIVO and 2.5x longer per step than RWS and IWAE. However, Figure 4 shows that FIVO, NASMC, IWAE, and RWS cannot match NAS-X’s performance even with 64 times more computation (256 particles). SIXO only matches NAS-X’s performance with 4x as many particles. Therefore, NAS-X uses computational resources much more effectively than other methods.

### Gradient variance and bias
Figure 5 (left) shows NAS-X attains lower variance gradient estimates than IWAE, FIVO, and SIXO with comparable variance to RWS. We also studied the bias in Figure 5 (middle) by approximating the true gradient using SMC with 256 particles and the best proposal found in the HH inference experiments. NAS-X's gradients are lower bias than all methods but FIVO, but FIVO's gradients are also the highest variance. We believe FIVO’s gradients appear less biased because its parameters are pushed towards degenerate values where gradient estimation is “easier”. We illustrate this in Figure 5 (right), where we plot LML bounds.

### New theoretical results
We present two theoretical results illustrating NAS-X’s advantages in idealized settings. One concerns the consistency of NAS-X’s gradients and the other concerns unbiasedness.

Proposition I (informal): For any proposal and the optimal twists, NAS-X’s gradient estimate of the log marginal likelihood converges almost surely to the true gradient as the number of particles approaches infinity. This is not true for NASMC.

Sketch: This follows from 1) SMC’s strongly consistent estimates of expectations of test functions with respect to a normalized target distribution (Theorem 7.4.3 Del Moral [2004]), 2) NAS-X’s use of smoothing distributions as the intermediate targets, and 3) the twist optimality.

This is not true for NASMC since it uses filtering SMC’s intermediate particle approximations (to avoid particle degeneracy), which approximate the filtering distributions and not the true posterior.

Proposition II (informal): For any finite number of particles, NAS-X’s gradient estimates are unbiased, provided that the proposal equals the true posterior and twists are optimal. This is not true for NASMC.

Sketch: Under the stated assumptions, both NAS-X and NASMC propose particles from the true posterior. However, the different intermediate target distributions will affect how these particles are distributed after reweighting. For NAS-X, the particles will have equal weight since they are reweighted using the smoothing targets. Thus, after reweighting, the particles are still samples from the true posterior. In contrast, in NASMC, the samples are reweighted by filtering targets and will be distributed according to the filtering distributions instead of the true posterior.

In the revision of this paper, we will include these statements and formal proofs.

### Limitations
NAS-X lacks a unified objective; the proposal and model are updated using different objectives. FIVO and SIXO have a unified objective for model and proposal. This didn’t have a practical impact on our experiments but could be relevant in other applications. NAS-X also does not work in online settings where future data is unavailable.

---

### Author Response · Authors · 2023-08-21

We thank the reviewers for engaging with us during the rebuttal period!

In response to the original reviews, we provided new results demonstrating strong improvements over several methods (NASMC, ELBO, IWAE, FIVO, SIXO, and RWS) on both synthetic (Gaussian SSM) and real-world problems (Hodgkin-Huxley). We also provided empirical and theoretical results on gradient variance and bias as well as wall-clock performance.

Regarding twist learning, we wanted to emphasize that twist learning was robust and easy in our experiments, even beyond the Gaussian SSM task. The classification accuracy generally reaches ~80-90% on the Hodgkin-Huxley tasks.

Regarding impact, we wanted to emphasize that NAS-X is a drop-in replacement for SIXO (requiring only minimal changes to code) but performs significantly better (as shown by our results) and can handle discrete latent variables, which SIXO cannot.

Thanks again for taking the time to review our paper and provide valuable feedback!

---

### Decision · Program_Chairs · 2023-09-21

**Decision:**

Accept (poster)

**Comment:**

Based on unanimous approval from all reviewers, this paper is accepted. The authors effectively addressed concerns raised during the review process and provided comprehensive responses during the rebuttal stage.

In their rebuttal, the authors present new results showcasing substantial advancements over various existing methods. Their introduced approach, NAS-X, exhibits superior performance compared to NASMC, ELBO, IWAE, FIVO, SIXO, and RWS on synthetic (Gaussian SSM) and real-world (Hodgkin-Huxley) problems. They further support their claims by offering empirical and theoretical evidence on gradient variance and bias, as well as the computational efficiency of their method.

Additionally, the authors emphasize the robustness and simplicity of twist learning in their experiments, which continues to be effective beyond just the Gaussian SSM task.

Furthermore, the authors highlight the practical benefits of NAS-X. It can readily replace SIXO, requiring only minimal code modifications, while significantly outperforming SIXO. Moreover, NAS-X allows for the handling of discrete latent variables, which SIXO fails to accomplish.

Overall, the paper's substantial improvements, backed by empirical and theoretical results, warrant its acceptance.